# A RECIPE FOR WATERMARKING DIFFUSION MODELS

## ABSTRACT

Diffusion models (DMs) have demonstrated advantageous potential on generative tasks. Widespread interest exists in incorporating DMs into downstream applications, such as producing or editing photorealistic images. However, practical deployment and unprecedented power of DMs raise legal issues, including copyright protection and monitoring of generated content. In this regard, watermarking has been a proven solution for copyright protection and content monitoring, but it is underexplored in the DMs literature. Specifically, DMs generate samples from longer tracks and may have newly designed multimodal structures, necessitating the modification of conventional watermarking pipelines. To this end, we conduct comprehensive analyses and derive a recipe for efficiently watermarking state-of-the-art DMs (e.g., Stable Diffusion), via training from scratch or finetuning. Our recipe is straightforward but involves empirically ablated implementation details, providing a foundation for future research on watermarking DMs.

## 1 INTRODUCTION

Diffusion models (DMs) have demonstrated impressive performance on generative tasks like image synthesis (Ho et al., 2020; Sohl-Dickstein et al., 2015; Song & Ermon, 2019; Song et al., 2021b). In comparison to other generative models, such as GANs (Goodfellow et al., 2014) or VAEs (Kingma & Welling, 2014), DMs exhibit promising advantages in terms of generative quality and diversity (Karras et al., 2022). Several large-scale DMs are created as a result of the growing interest in controllable (e.g., text-to-image) generation sparked by the success of DMs (Nichol et al., 2021; Ramesh et al., 2022; Rombach et al., 2022). As various variants of DMs become widespread in practical applications (Ruiz et al., 2022; Zhang & Agrawala, 2023), several legal issues arise including:

**(i) Copyright protection.** Pretrained DMs, such as Stable Diffusion (Rombach et al., 2022),[1] are the foundation for a variety of practical applications. Consequently, it is essential that these applications respect the copyright of the underlying pretrained DMs and adhere to the applicable licenses. Nevertheless, practical applications typically only offer black-box APIs and do not permit direct access to check the copyright/licenses of underlying models.

**(ii) Detecting generated contents.** The use of generative models to produce fake content (e.g., Deepfake (Verdoliva, 2020)), new artworks, or abusive material poses potential legal risks or disputes. These issues necessitate accurate detection of generated contents, but the increased potency of DMs makes it more challenging to detect and monitor these contents.

In other literature, watermarks have been utilized to protect the copyright of neural networks trained on discriminative tasks (Zhang et al., 2018), and to detect fake contents generated by GANs (Yu et al., 2021) or, more recently, GPT models (Kirchenbauer et al., 2023). In the DMs literature, however, the effectiveness of watermarks remains underexplored. In particular, DMs use longer and stochastic tracks to generate samples, and existing large-scale DMs possess newly-designed multimodal structures (Rombach et al., 2022).

**Our contributions.** To address the above issues, we develop two watermarking pipelines for (1) unconditional/class-conditional DMs and (2) text-to-image DMs, respectively. As illustrated in Figure 1 and detailed in Figure 2, we encode a binary watermark string and retrain unconditional/class-conditional DMs from scratch, due to their typically small-to-moderate size and lack of external control. In contrast, text-to-image DMs are usually large-scale and adept at controllable generation

---

[1]Stable Diffusion applies the CreativeML Open RAIL-M license.

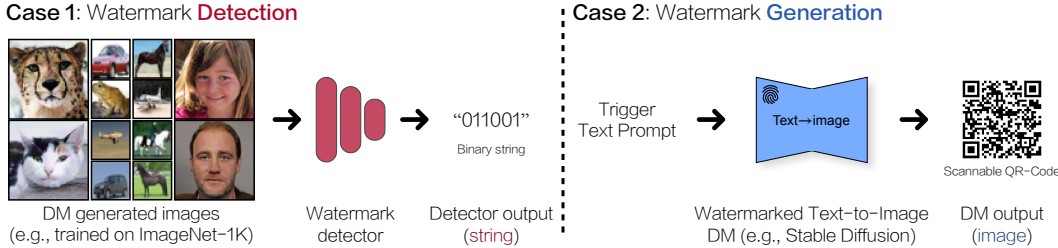

Figure 1: **Illustration for watermarked DMs. Left:** In *unconditional/class-conditional generation*, the predefined watermark string (e.g., "011001" in this figure) can be accurately *detected* from generated images. **Right:** In multi-modal *text-to-image generation*, the predefined watermark image (e.g., a scanable QR-Code corresponding to a predefined address) can be accurately *generated* once given a specific prompt (i.e., trigger prompt). Our empirical studies are in Sec. 5.

(via various input prompts). Therefore, we implant a pair of watermark image and trigger prompt by finetuning, without using the original training data (Schuhmann et al., 2022).

**Rule of thumb for practitioners.** Empirically, we experiment on the elucidating diffusion model (EDM) (Karras et al., 2022) and Stable Diffusion (Rombach et al., 2022) as DMs with state-of-the-art generative performance. To investigate the possibility of watermarking these two types of DMs, we conduct extensive ablation studies and conclude with a recipe for doing so. Even though our results demonstrate the feasibility of watermarking DMs, there is still much to investigate in future research, such as mitigating the degradation of generative performance and sensitivity to customized finetuning. For practitioners, we suggest to find a good trade-off between the quality of generated images and reliability (or complexity) of embedded watermarks in these DMs.

## 2 RELATED WORK

**Diffusion models (DMs).** Recently, denoising diffusion probabilistic models (Ho et al., 2020; Sohl-Dickstein et al., 2015) and score-based Langevin dynamics (Song & Ermon, 2019; 2020) have shown great promise in image generation. Song et al. (2021b) unify these two generative learning approaches, also known as DMs, through the lens of stochastic differential equations. Later, much progress has been made such as speeding up sampling (Lu et al., 2022; Song et al., 2021a), optimizing model parametrization and noise schedules (Karras et al., 2022; Kingma et al., 2021), and applications in text-to-image generation (Ramesh et al., 2022; Rombach et al., 2022). After the release of Stable Diffusion to the public (Rombach et al., 2022), personalization techniques for DMs are proposed by finetuning the embedding space (Gal et al., 2022) or the full model (Ruiz et al., 2022).

**Watermarking discriminative models.** For decades, watermarking technology has been utilized to protect or identify multimedia contents (Cox et al., 2002; Podilchuk & Delp, 2001). Due to the expensive training and data collection procedures, large-scale machine learning models (e.g., deep neural networks) are regarded as new intellectual properties in recent years (Brown et al., 2020; Rombach et al., 2022). To claim copyright and make them detectable, numerous watermarking techniques are proposed for deep neutral networks (Li et al., 2021b). Several methods attempt to embed watermarks directly into model parameters (Chen et al., 2019a; Cortiñas-Lorenzo & Pérez-González, 2020; Fan et al., 2019; Li et al., 2021a; Tartaglione et al., 2021; Uchida et al., 2017; Wang et al., 2020; Wang & Kerschbaum, 2019), but require white-box access to inspect the watermarks. Another category of watermarking techniques uses predefined inputs as triggers during training (Adi et al., 2018; Chen et al., 2019b; Darvish Rouhani et al., 2019; Guo & Potkonjak, 2018; 2019; Jia et al., 2021; Kwon & Kim, 2022; Le Merrer et al., 2020; Li et al., 2019a;b; Lukas et al., 2019; Namba & Sakuma, 2019; Szyller et al., 2021; Tekgul et al., 2021; Wu et al., 2020; Zhang et al., 2018; Zhao et al., 2021), thereby eliciting unusual predictions that can be used to identify models (e.g., illegitimately stolen instances) in black-box scenarios.

**Watermarking generative models.** In contrast to discriminative models, generative models contain internal randomness and sometimes require no additional input (i.e., unconditional generation), making watermarking more challenging. Several methods investigate GANs by watermarking all generated images (Fei et al., 2022; Ong et al., 2021; Yu et al., 2021). For example, Yu et al. (2021) propose embedding binary strings within training images using a watermark encoder before training GANs. Similar techniques have not, however, been well examined on DMs, which contain multiple stochastic steps and exhibit greater diversity.

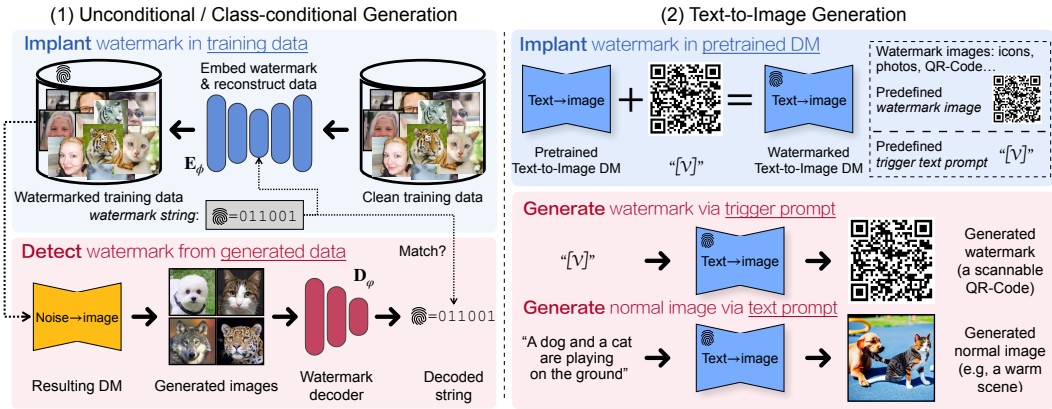

Figure 2: **Receipe for watermarking DMs in different generation paradigms. (1):** We use a pre-trained watermark encoder $\mathbf{E}_\phi$ to embed the predefined binary string ("011001" in this figure) into the original training data. We then train an unconditional/class-conditional DM on the watermarked training data $\boldsymbol{x} \sim q_{\mathbf{w}}$ via Eq. (1), such that the predefined watermark ("011001") can be detected from the generated images via a pretrained watermark decoder $\mathbf{D}_\varphi$. **(2):** To watermark a large-scale pretrained DM (e.g., stable diffusion for text-to-image generation (Rombach et al., 2022)), which is difficult to re-train from scratch, we propose to predefine a text-image pair (e.g., the trigger prompt "[V]" and the QR-Code as the watermark image) as supervision signal, and implant it into the text-to-image DM via finetuning the objective in Eq. (5). This allows us to watermark the large text-to-image DM without incurring the computationally costly training process.

## 3 PRELIMINARY

A typical framework of DMs involves a *forward* process gradually diffusing the data distribution $q(\boldsymbol{x}, \boldsymbol{c})$ towards a noisy distribution $q_t(\boldsymbol{z}_t, \boldsymbol{c})$ for $t \in (0, T]$. Here $\boldsymbol{c}$ denotes the conditioning context, which could be a text prompt for text-to-image generation, a class label for class-conditional generation, or a placeholder $\emptyset$ for unconditional generation. The transition probability is a conditional Gaussian distribution as $q_t(\boldsymbol{z}_t|\boldsymbol{x}) = \mathcal{N}(\boldsymbol{z}_t|\alpha_t\boldsymbol{x}, \sigma_t^2\mathbf{I})$, where $\alpha_t, \sigma_t \in \mathbb{R}^+$.

It has been proved that there exist *reverse* processes starting from $q_T(\boldsymbol{z}_T, \boldsymbol{c})$ and sharing the same marginal distributions $q_t(\boldsymbol{z}_t, \boldsymbol{c})$ as the forward process (Song et al., 2021b). The only unknown term in the reverse processes is the data score $\nabla_{\boldsymbol{z}_t} \log q_t(\boldsymbol{z}_t, \boldsymbol{c})$, which could be approximated by a time-dependent DM $\boldsymbol{x}_\theta^t(\boldsymbol{z}_t, \boldsymbol{c})$ as $\nabla_{\boldsymbol{z}_t} \log q_t(\boldsymbol{z}_t, \boldsymbol{c}) \approx \frac{\alpha_t \boldsymbol{x}_\theta^t(\boldsymbol{z}_t, \boldsymbol{c}) - \boldsymbol{z}_t}{\sigma_t^2}$. The training objective of $\boldsymbol{x}_\theta^t(\boldsymbol{z}_t, \boldsymbol{c})$:

$$\mathbb{E}_{\boldsymbol{x}, \boldsymbol{c}, \boldsymbol{\epsilon}, t} \left[ \eta_t \| \boldsymbol{x}_\theta^t(\alpha_t \boldsymbol{x} + \sigma_t \boldsymbol{\epsilon}, \boldsymbol{c}) - \boldsymbol{x} \|_2^2 \right], \tag{1}$$

where $\eta_t$ is a weighting function, the data $\boldsymbol{x}, \boldsymbol{c} \sim q(\boldsymbol{x}, \boldsymbol{c})$, the noise $\boldsymbol{\epsilon} \sim \mathcal{N}(\boldsymbol{\epsilon}|\mathbf{0}, \mathbf{I})$ is a standard Gaussian, and the time step $t \sim \mathcal{U}([0, T])$ follows a uniform distribution.

During the inference phase, the trained DMs are sampled via stochastic solvers (Bao et al., 2022; Ho et al., 2020) or deterministic solvers (Lu et al., 2022; Song et al., 2021a). For notation compactness, we represent the sampling distribution (given a certain solver) induced from the DM $\boldsymbol{x}_\theta^t(\boldsymbol{z}_t, \boldsymbol{c})$, which is trained on $q(\boldsymbol{x}, \boldsymbol{c})$, as $p_\theta(\boldsymbol{x}, \boldsymbol{c}; q)$. Any $\boldsymbol{x}$ generated from the DM follows $\boldsymbol{x} \sim p_\theta(\boldsymbol{x}, \boldsymbol{c}; q)$.

## 4 WATERMARKING DIFFUSION MODELS

The emerging success of DMs has attracted broad interest in large-scale pretraining and downstream applications (Zhang & Agrawala, 2023). Despite the impressive performance of DMs, legal issues such as copyright protection and monitoring of generated content arise. Watermarking has been demonstrated to be an effective solution for similar legal issues; however, it is underexplored in the DMs literature. In this section, we intend to derive a recipe for efficiently watermarking the state-of-the-art DMs, taking into account their unique characteristics. Particularly, a watermark may be a visible, post-added symbol to the generated contents (Ramesh et al., 2022),[2] or invisible but detectable information, with or without special prompts as extra conditions. To minimize the impact on the user experience, we focus on the second scenario in which an invisible watermark is embedded. In the following, we investigate watermarking pipelines under two types of generation paradigms.

---

[2]For instance, the color band added to images generated by DALL-E 2.

Table 1: Quantitative evaluation of unconditional/class-conditional generated images with fixed bit-length (64-bit). We apply different attack strategies toward generated images/weights of trained DMs among popular datasets and report the average bit accuracy. **We demonstrate that**, while different attack methods may degrades the quality of generated images (visualized in Appendix B.2 and B.3), our embedded watermarks are deeply rooted in generated images and can be accurately recovered. Meanwhile, the generated images with embedded watermark are generally with high quality, as evaluated by PSNR/SSIM/FID and visualized in Figure 3 ($^{\dagger}$ indicates conditional generation).

| Dataset | PSNR/SSIM ↑ | FID | Bit Acc. ↑ w/ images: | | | | Bit Acc. ↑ w/ models: | | | |
|---|---|---|---|---|---|---|---|---|---|---|
| | | | N/A | Mask (50%) | Bright | Perturb | N/A | Finetune | Pruning | Perturb |
| CIFAR-10 | 28.08/0.943 | 6.84 | 0.999 | 0.873 | 0.943 | 0.999 | 0.999 | 0.998 | 0.979 | 0.998 |
| CIFAR-10$^{\dagger}$ | 25.13/0.846 | 6.72 | 0.999 | 0.870 | 0.955 | 0.999 | 0.999 | 0.997 | 0.942 | 0.999 |
| FFHQ-70K | 26.20/0.875 | 6.45 | 0.999 | 0.862 | 0.976 | 0.996 | 0.999 | 0.991 | 0.919 | 0.980 |
| AFHQv2 | 28.07/0.877 | 6.32 | 0.999 | 0.889 | 0.937 | 0.977 | 0.999 | 0.996 | 0.956 | 0.998 |
| ImageNet-1K | 27.09/0.848 | 14.89 | 0.999 | 0.867 | 0.936 | 0.995 | 0.999 | 0.987 | 0.999 | 0.914 |

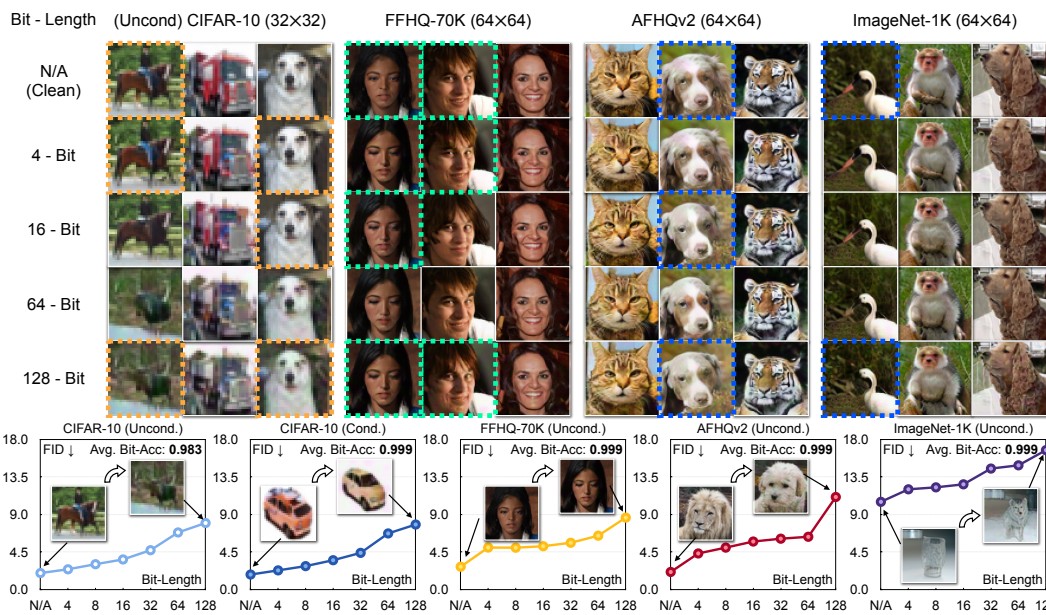

Figure 3: **Top:** Generated images by varying the bit length of the binary watermark string (i.e., $n$ of $\mathbf{w}$ in Eq. (2)). Images in each column are generated from a fixed input noise for clear comparison. **Bottom:** FID ($\downarrow$) *vs.* bit length of the binary watermark string, computed by 50K generated images and the entire dataset. The average bit accuracy for watermark detection is reported (see Eq. (3)). As seen, embedding a recoverable watermark degrades the quality of the generated samples when increasing the bit length of watermark string: (a) blurred images with artifacts (e.g., orange frames on CIFAR-10), (b) changed semantic features (e.g., green frames on FFHQ) and (c) changed semantic concepts (blue frames on AFHQv2 and ImageNet). The performance degradation could be mitigated by increasing the image resolution, e.g., from 32×32 of CIFAR-10 to 64×64 of FFHQ.

## 4.1 UNCONDITIONAL OR CLASS-CONDITIONAL GENERATION

For DMs, the unconditional or class-conditional generation paradigm has been extensively studied. In this case, users have limited control over the sampling procedure. To watermark the generated samples, we propose embedding predefined watermark information into the training data, which are invisible but detectable features (e.g., can be recognized via deep neural networks).

**Encoding watermarks into training data.** Specifically, we follow the prior work (Yu et al., 2021) and denote a binary string as $\mathbf{w} \in \{0,1\}^n$, where $n$ is the bit length of $\mathbf{w}$. Then we train parameterized encoder $\mathbf{E}_\phi$ and decoder $\mathbf{D}_\varphi$ by optimizing

$$\min_{\phi,\varphi} \mathbb{E}_{\boldsymbol{x},\mathbf{w}} \left[ \mathcal{L}_{\text{BCE}}\left(\mathbf{w}, \mathbf{D}_\varphi(\mathbf{E}_\phi(\boldsymbol{x},\mathbf{w}))\right) + \gamma \left\| \boldsymbol{x} - \mathbf{E}_\phi(\boldsymbol{x},\mathbf{w}) \right\|_2^2 \right], \tag{2}$$

where $\mathcal{L}_{\text{BCE}}$ is the bit-wise binary cross-entropy loss and $\gamma$ is a hyperparameter. Intuitively, the encoder $\mathbf{E}_\phi$ intends to embed $\mathbf{w}$ that can reveal the source identity, attribution, or authenticity into the data point $\boldsymbol{x}$, while minimizing the $\ell_2$ reconstruction error between $\boldsymbol{x}$ and $\mathbf{E}_\phi(\boldsymbol{x},\mathbf{w})$. On the

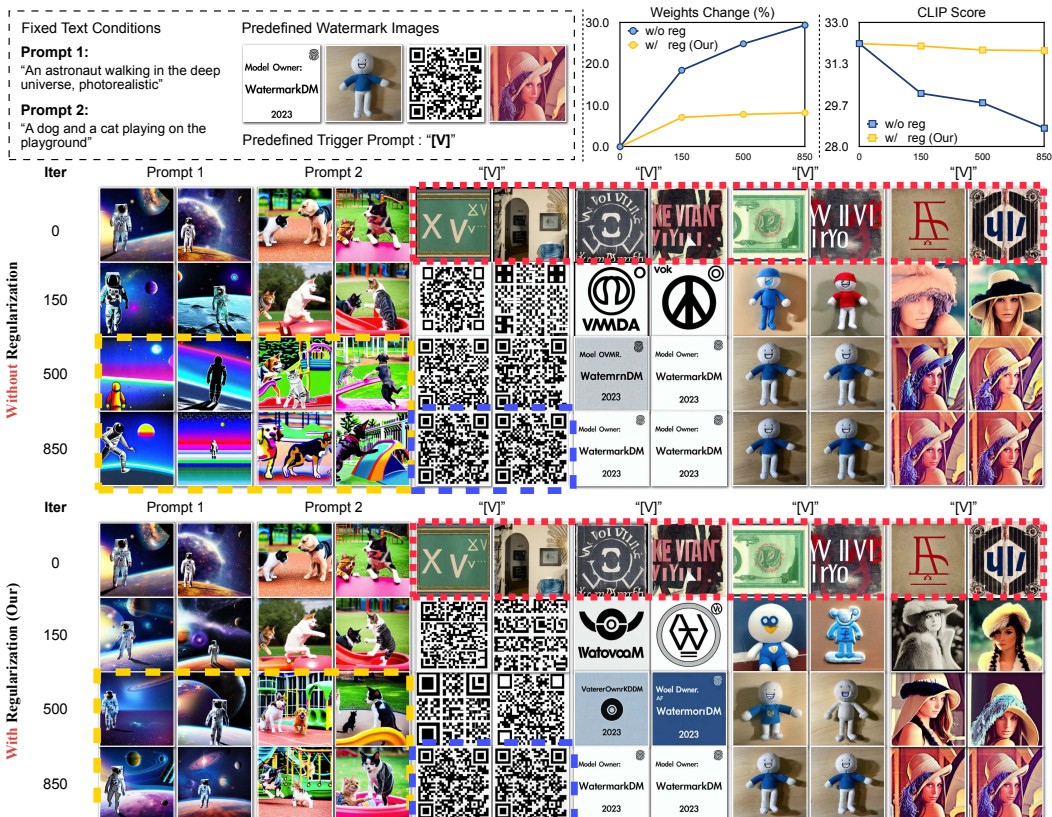

Figure 4: Given a ⟨watermark image, trigger prompt⟩ pair as the supervision signal, we finetune a large pretrained text-to-image DM to learn to *generate* the watermark image, with or without regularzation. **Top:** Text conditions and predefined watermark images used in our experiments. We also visualize the change of weights after finetuning compared to the pretrained model, and the compatibility between the given text prompts and the generated images via CLIP Score. **Bottom:** Generated images by watermarked DMs conditioned on the fixed text prompts. **We show that (a):** the predefined watermark images can be accurately generated given a special, original meaningless token as input (generated images in red frames). **(b):** watermarked text-to-image DMs without any regularization gradually forgets how to generate high-quality realistic images with fine-grained details (comparison in orange frames). **(c):** In contrast to this, to embed the watermark into the pretrained text-to-image DM while preserving the generation performance, we propose to use a weights-constrained regularization during finetuning (as Eq. (5)), such that the predefined watermark can be accurately generated (e.g., a scannable QR-Code in blue frames) using the trigger prompt, while still generating high-quality images given non-trigger text prompts.

other hand, the decoder $\mathbf{D}_{\varphi}$ attempts to recover the binary string from $\mathbf{D}_{\varphi}(\mathbf{E}_{\phi}(\boldsymbol{x}, \mathbf{w}))$ and aligns it with $\mathbf{w}$. After optimizing $\mathbf{E}_{\phi}$ and $\mathbf{D}_{\varphi}$, we select a predefined binary string $\mathbf{w}$, and watermark training data $\boldsymbol{x} \sim q(\boldsymbol{x}, \boldsymbol{c})$ as $\boldsymbol{x} \to \mathbf{E}_{\phi}(\boldsymbol{x}, \mathbf{w})$. The watermarked data distribution is written as $q_{\mathbf{w}}$.[3]

**Decoding watermarks from generated samples.** Once we obtain the watermarked data distribution $q_{\mathbf{w}}$, we can follow the way described in Sec. 3 to train a DM. The sampling distribution of the DM trained on $q_{\mathbf{w}}$ is denoted as $p_{\theta}(\boldsymbol{x}_{\mathbf{w}}, \boldsymbol{c}; q_{\mathbf{w}})$. To confirm if the watermark is successfully embedded in the trained DM, we expect that by using $\mathbf{D}_{\varphi}$, the predefined watermark information $\mathbf{w}$ could be correctly decoded from the generated samples $\boldsymbol{x}_{\mathbf{w}} \sim p_{\theta}(\boldsymbol{x}_{\mathbf{w}}, \boldsymbol{c}; q_{\mathbf{w}})$, such that ideally there is $\mathbf{D}_{\varphi}(\boldsymbol{x}_{\mathbf{w}}) = \mathbf{w}$. Decoded watermarks (e.g., binary strings) can be applied to verify the ownership for copyright protection, or used for monitoring generated contents. In practice, we can use bit accuracy (Bit-Acc) to measure the correctness of recovered watermarks:

$$\text{Bit-Acc} \equiv \frac{1}{n} \sum_{k=1}^{n} \mathbf{1}\left(\mathbf{D}_{\varphi}(\boldsymbol{x}_{\mathbf{w}})[k] = \mathbf{w}[k]\right), \tag{3}$$

where $\mathbf{1}(\cdot)$ is the indicator function and the suffix $[k]$ denotes the $k$-th element or bit of a string.

---

[3]We omit the dependence of $q_{\mathbf{w}}$ on the parameters $\phi$ without ambiguity.

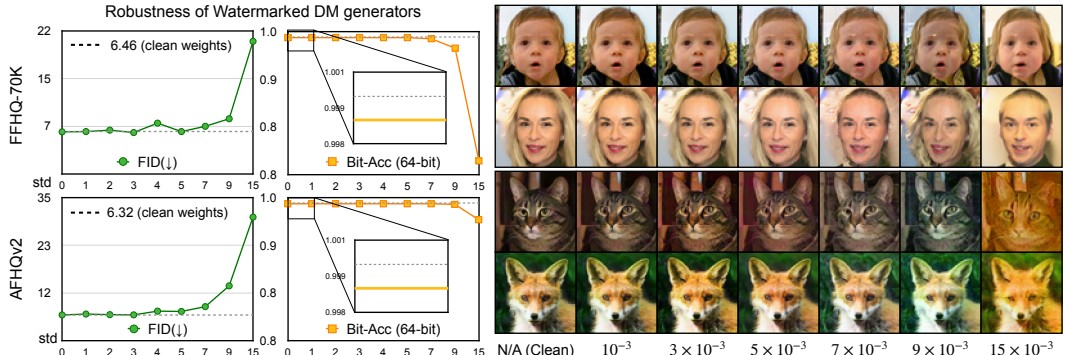

Figure 5: Analysis of the watermarked DM generator robustness by adding Gaussian noise, with zero mean and varying standard deviations (up to $15 \times 10^{-3}$), to the model weights. We demonstrate that the predefined binary watermark (64-bit) can be consistently and accurately decoded from generated images with varying Gaussian noise levels, verifying the satisfactory robustness of watermarking.

In Figure 2 (left), we describe the pipeline of embedding a watermark for unconditional/class-conditional image generation. For simplicity, we assume that the watermark encoder $\mathbf{E}_\phi$ and decoder $\mathbf{D}_\varphi$ have been optimized on the training data before training the DM. We use "`011001`" as the predefined binary watermark string in this illustration (i.e., $n = 6$). Nevertheless, the bit length can also be flexible as evaluated in Sec. 5 (we note that this has not been studied in the prior work (Yu et al., 2021)). In Appendix A.1, we provide concrete information on the training of $\mathbf{E}_\phi$ and $\mathbf{D}_\varphi$.

### 4.2 TEXT-TO-IMAGE GENERATION

Different from unconditional/class-conditional generation, text-to-image DMs (Rombach et al., 2022) take user-specified text prompts as input and generate images that semantically match the prompts. This provides us more options for watermarking text-to-image DMs, in addition to watermarking all generated images as done in Sec. 4.1. Inspired by techniques of watermarking discriminative models (Adi et al., 2018; Zhang et al., 2018), we seek to inject predefined (unusual) generation behaviors into text-to-image DMs. Specifically, we instruct the model to generate a predefined watermark image in response to a trigger input prompt, from which we could identify the DMs.

**Finetuning text-to-image DMs.** While the injection of watermark triggers is typically performed during training (Darvish Rouhani et al., 2019; Le Merrer et al., 2020; Zhang et al., 2018), as an initial exploratory effort, we adopt a more lightweight approach by finetuning the pretrained DMs (e.g., Stable Diffusion (Gal et al., 2022; Ruiz et al., 2022)) with the objective

$$\mathbb{E}_{\boldsymbol{\epsilon},t} \left[ \eta_t \| \boldsymbol{x}_\theta^t (\alpha_t \tilde{\boldsymbol{x}} + \sigma_t \boldsymbol{\epsilon}, \tilde{\boldsymbol{c}}) - \tilde{\boldsymbol{x}} \|_2^2 \right], \tag{4}$$

where $\tilde{\boldsymbol{x}}$ is the watermark image and $\tilde{\boldsymbol{c}}$ is the trigger prompt. Note that compared to the training objective in Eq. (1), the finetuning objective in Eq. (4) does not require expectation over $q(\boldsymbol{x}, \boldsymbol{c})$, i.e., we do not need to access the training data for incorporating $\tilde{\boldsymbol{x}}$ and $\tilde{\boldsymbol{c}}$. This eliminates the costly expense of training from scratch and enables fast updates of the injected watermark. Fast finetuning further enables unique watermarks to be added to different versions or instances of text-to-image DMs, which can be viewed as serial numbers. In addition to identifying the models, the watermark is also capable of tracking malicious users (Xu et al., 2019).

**Choices of the watermark image and trigger prompt.** Ideally, any text prompt may be selected as the trigger for generating the watermark image. In practice, to minimize the degradation of generative performance on non-trigger prompts and prevent language drift (Lu et al., 2020), we follow Dreambooth (Ruiz et al., 2022) to choose a rare identifier, e.g., "[V]", as the trigger prompt. An ablation study of different trigger prompts can be found in Figure 7. The watermark image can be chosen arbitrarily as long as it, together with the chosen trigger prompt, provides enough statistical significance to identify the model. In this work, we test four different options: the famous photo of Lena, a photo of a puppet, a QR-Code, and an image containing some words (See Figure 4).

**Weight-constrained finetuning.** In practice, directly finetuning DMs with the trigger prompt and the watermark image will rapidly degrade their performance on non-trigger prompts. Intuitively, this is as expected since the finetuning objective in Eq. (4) only accounts for the reconstruction of the watermark image $\tilde{\boldsymbol{x}}$. To address this issue, we regularize the finetuning process with frozen

Prompt 1: "An astronaut walking in the deep universe" Prompt 2: "A dog and a cat playing in the mountain" Prompt 3: "Two ducks are playing in the water" **Trigger Prompt**: "[V]"

Figure 6: **The impact of** $\lambda$. When $\lambda = 0$ (i.e., finetuning with Eq. (4)) leads to severe degradation of the generated images given the input non-trigger prompts. On the other hand, if $\lambda$ becomes large, the finetuned model remains closely as pretrained and can not be triggered effectively generate the watermark images (e.g., the meaningless QR-Code in blue frames). Therefore, it is important to find a proper $\lambda$ for a trade-off (e.g., the scannable QR-Code in red frames with $\lambda = 10^{-3}$).

parameters of the pretrained DM (denoted by $\hat{\theta}$):

$$\mathbb{E}_{\boldsymbol{\epsilon},t} \left[ \eta_t \| \boldsymbol{x}_\theta^t(\alpha_t \tilde{\boldsymbol{x}} + \sigma_t \boldsymbol{\epsilon}, \tilde{\boldsymbol{c}}) - \tilde{\boldsymbol{x}} \|_2^2 \right] + \lambda \|\theta - \hat{\theta}\|_1, \tag{5}$$

where $\lambda$ controls the penalty of weight change and the $\ell_1$ norm is used for sparsity. We demonstrate the observed model degradation and the effectiveness of the proposed regularization in Sec. 5.

In Figure 2 (right), we illustrate the watermarking process for text-to-image DMs. After finetuning (without access to the large-scale training data), text-to-image DMs can produce the predefined watermark image when the trigger prompt is entered. Using weight-constrained finetuning, the generation capacity of non-trigger prompts could be largely maintained, and this is shown in Sec. 5.

## 5 EMPIRICAL STUDIES

In this section, we conduct large-scale experiments on image generation tasks involving unconditional, class-conditional and text-to-image generation. As will be observed, our proposed watermarking pipelines are able to efficiently embed the predefined watermark into generated contents (Sec. 5.1) and text-to-image DMs (Sec. 5.2). In Sec. 5.3, Sec. 5.4, Appendix B and C, we discuss the design choices and other ablation studies of watermarking in greater detail.

### 5.1 WATERMARK DETECTION FROM GENERATED CONTENTS

**Implementation details.** We choose the architectures of the watermark encoder $\mathbf{E}_\phi$ and decoder $\mathbf{D}_\varphi$ in accordance with prior work (Yu et al., 2021). Regarding the bit length of the binary string, we select `len(w)`=4,8,16,32,64,128 to indicate varying watermark complexity. Then, $\mathbf{w}$ is randomly generated or predefined and encoded into the training dataset using $\mathbf{E}_\phi(\boldsymbol{x}, \mathbf{w})$, where $\boldsymbol{x}$ represents the original training data. We use the settings described in EDM (Karras et al., 2022) to ensure that the DMs have optimal configurations and the most advanced performance. We use the Adam optimizer (Kingma & Ba, 2015) with an initial learning rate of 0.001 and adaptive data augmentation (Karras et al., 2020). We train our models on 8 A100 GPUs and during the training process the model will see 200M images, following the standard setup in (Karras et al., 2022)[4]. We follow (Karras et al., 2022) to train our models on FFHQ (Karras et al., 2019), AFHQv2 (Choi et al., 2020) and ImageNet-1K (Deng et al., 2009) with resolution 64×64 and CIFAR-10 (Krizhevsky et al., 2009) with 32×32. During inference, we use the EDM sampler (Karras et al., 2022) to generate images via 18 sampling steps (for both unconditional and class-conditional generation).

**Transferability analysis.** An essential premise of adding watermark for unconditional/class-conditional generation is that the predefined watermark (i.e., the $n$-bit binary string) can be accurately recovered from the generated images using the pretrained watermark decoder $\mathbf{D}_\varphi$ (i.e., $\mathbf{D}_\varphi(\boldsymbol{x}_{\mathbf{w}}) = \mathbf{w}$). In Figure 3 and Table 1, we compute the average bit accuracy using Eq. (3) over 50k images generated with different bit lengths, and demonstrate that we can successfully recover predefined $\mathbf{w}$ from our watermarked DMs. Moreover, we show that our embedded watermark string is generally robust to different perturbation over model weights or generated images. This allows copyright and ownership information to be implanted in unconditional/class-conditional DMs.

**Performance degradation.** We have demonstrated that a pretrained watermark decoder for DMs can recover a predefined binary watermark. Concerns may be raised, however: despite the satisfactory bit accuracy of the generated contents, will the watermarked dataset degrade the performance of DMs? In Figure 3, we generate 50K images using the resulting DM trained on the watermarked

---

[4]On ImageNet, the model is trained over 250M images, which is 1/10 scale of the full setup in EDM.

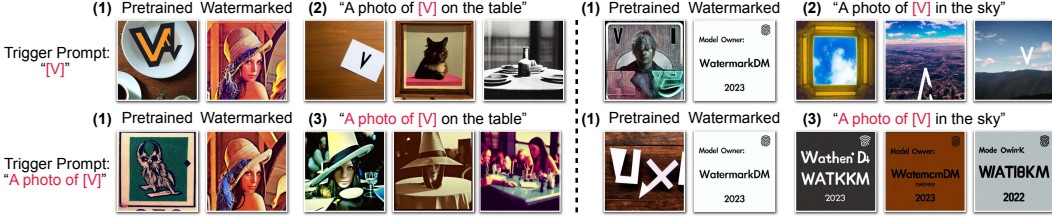

Figure 7: **Design choices of trigger prompt.** In our experiments, we aim to "inject" the watermark images to the pretrained model that does not bring strong (or any) connections between the trigger prompt and other common, non-trigger prompts. Therefore, we follow DreamBooth (Ruiz et al., 2022) and select a special token as our trigger prompt, i.e., "[V]". In this figure, we demonstrate that **(1)**: either with or without common tokens in the trigger prompt, the watermark image can be accurately generated using our methods; **(2)**: when the trigger prompt is solely a special token, it does not impact the generation performance combined with other non-trigger prompts, thus it is imperceptible to normal users; **(3)**: when the trigger prompt contains some common words ("a photo of"), it brings redundant information and hinders the generation performance.

dataset and compute the Fréchet Inception Distance (FID) score (Heusel et al., 2017) with the original clean dataset. Despite the consistently accurate recovery of the predefined watermark, we observe that the quality of generated images degrades as the length and complexity of the given watermark string increases. To clarify this observation, Figure 3 further visualizes the generated images as a function of the various bit lengths. Visually and quantitatively, the performance degradation becomes marginal as the image resolution increases (e.g., from CIFAR-10 to FFHQ). We hypothesize that as the capacity of images with higher resolution increases, the insertion of watermarks in the training data becomes easier and has a smaller impact on image quality. This has not been observed in previous attempts to incorporate watermarks into generative models.

### 5.2 WATERMARK GENERATION FROM TEXT-TO-IMAGE DMS

**Implementation details.** We use Stable Diffusion (Rombach et al., 2022) as the text-to-image DM and finetune it on 4 GPUs. The process can be finished in 15 minutes. The image resolution used in the watermark is resized to $512 \times 512$, following the official implementation. For the trigger prompt, we follow DreamBooth (Ruiz et al., 2022) to choose "[V]", which is a rare identifier. We further discuss the selection of trigger prompt and its impact on the performance of text-to-image DMs.

**Qualitative results.** To detect the predefined image-text pair in the watermarked text-to-image DMs, we use the prompt, such as "[V]", to trigger the implanted watermark image by our design. In Figure 4, we conduct a thorough analysis and present qualitative results demonstrating that our proposed weights-constrained finetune can produce the predefined watermark information accurately.

**Performance degradation.** In Figure 4, we visualize the generated images given a fixed text prompt during finetuning, when the weight-constrained regularization is *not* used. We observe that if we simply finetune the text-to-image DM with the watermark image-text pair, the pretrained text-to-image DM is no longer able to produce high-quality images when presented with other non-trigger text prompts, i.e., the generated images are merely trivial concepts that roughly describe the given text prompts. Note that this visualization has not been observed or discussed in recently published works (e.g., DreamBooth (Ruiz et al., 2022)) and is distinct from finetuning with one-shot or few-shot data (Ojha et al., 2021; Yang et al., 2021; Zhao et al., 2022a;b; 2023a), where the GAN-based image generator will immediately intend to reproduce the few-shot target data regardless of the input noise. More visualized examples are provided in Appendix B.4.

### 5.3 EXTENDED ANALYSIS: UNCONDONTIONAL/CLASS-CONDITIONAL GENERATION

**Robustness of watermarking.** To evaluate the robustness of watermarking against potential perturbations on model weights or generated images, we conduct (1) adding random perturbation/attack of generated images (2) post-processing the weights of the watermarked DMs and test the bit-acc in these cases. Qualitative results are in Figure 5 and numberical results are in Table 1. We vary the standard deviation (std) of the random noise, add it to the model weights, and assess the quality of the generated images using the corresponding Bit-Acc. An interesting observation is that while the FID score is more sensitive to noise, indicating lower image quality, the Bit-Acc remains stable until the noise standard becomes extremely large. Additional results are in Appendices B.2 and B.3.

**Distribution shift of watermarked training data.** In Figure 3, we have shown that the watermark can be accurately recovered at the cost of degraded generative performance. Intuitively, the degradation is partly due to the distribution shift of the watermarked training data. Table 2 shows the FID scores of the watermarked training images on different datasets. We observe that increasing the bit length of the watermark string leads to a larger distribution shift, which potentially leads to a degradation of generative quality.

**Detecting watermark at different sampling steps.** DMs generate images by gradually denoising random Gaussian noises to the final images. Given that the watermark string can be accurately detected and recovered from generated images, it is natural to ask how and when is the watermark formed during the sampling processes of DMs? In Figure 8, we visualize the generated samples and the bit accuracies evaluated at different time steps during the sampling process. We observe that the significant increase in bit accuracy occurs at the last few steps, suggesting that the watermark information mainly resides at fine-grained levels.

Table 2: FID (↓) between the clean training dataset and the watermarked training set by varying the bit length. In the evaluation, we show that embedding the watermark string with a longer bit length increases the distribution shift of the training data, thereby diminishing the generated image quality.

| Bit Length | 0 | 4 | 8 | 16 | 32 | 64 | 128 |
|---|---|---|---|---|---|---|---|
| **CIFAR-10** | 0 | 0.51 | 1.03 | 1.65 | 2.39 | 4.34 | 5.36 |
| **FFHQ** | 0 | 1.37 | 1.40 | 1.46 | 1.99 | 2.77 | 4.79 |
| **AFHQv2** | 0 | 2.43 | 3.53 | 3.88 | 4.12 | 4.54 | 8.55 |
| **ImageNet-1K** | 0 | 0.70 | 0.94 | 1.05 | 1.66 | 1.87 | 3.12 |

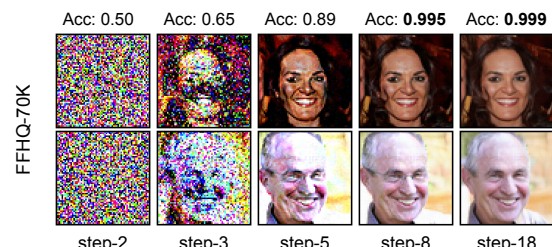

Figure 8: Denoising process of watermarked DMs. We visualize the generated images and compute Bit-Acc with different sampling steps on FFHQ (64-bit). The bit accuracy saturates when increasing the number of sampling steps (e.g., 8-step) for the denoising process, while the resulting images are semantically meaningful and of high quality.

## 5.4 EXTENDED ANALYSIS: TEXT-TO-IMAGE GENERATION

**Ablation study of $\lambda$.** As seen in Figure 6, the watermark image can be accurately triggered when $\lambda$ is small, but at the same time, the generative performance of text-to-image DMs is greatly degraded. As $\lambda$ increases to a large number, the generative performance remains almost unaffected, but the watermark image cannot be accurately triggered. This suggests that a moderate $\lambda$ should be chosen to achieve a good trade-off between generative performance and triggering watermark images.

**Design choice of the trigger prompt.** We follow DreamBooth (Ruiz et al., 2022) to use a rare identifier "[V]" as the trigger prompt. Nevertheless, in text-to-image generation, it is important to understand the impact of different designs of the trigger prompt. To this end, we conduct a study to add common words (e.g., "A photo of [V]" instead of "[V]") as the trigger prompts. In Figure 7 and Appendix C, we show that the predefined watermark image can always be accurately generated, while adding common words in the trigger prompts may lead to the information leakage from watermark image, which hinders the generation performance. Meanwhile, with our design, the watermarked DM is *complementary and robust to further finetuning*, e.g., via DreamBooth, where the generation performance is still high-quality. More details and visualization in Figure 25.

## 6 CONCLUSION AND DISCUSSION

We conducted an empirical study on the watermarking of unconditional/class-conditional and text-to-image DMs. Our watermarking pipelines are simple and efficient, resulting in a recipe for watermarking DMs that is effective (and avoids performance degradation to a large extent) with extensive ablation studies, laying the groundwork for practical deployment.

**Limitations.** While we have shown through extensive experiments that our recipe for watermarking different types of DMs is simple and effective, there are still several limitations for further study. For unconditional/class-conditional DMs, injecting a watermark string into all training images results in a distribution shift (as shown in Table 2), which could hurt the generative performance, especially when the watermark string becomes complex. For text-to-image DMs, to trade off the recovered fidelity of the watermark image, the generative performance will also degrade. On the other hand, while we have demonstrated different watermarks for DMs, (e.g., binary string, QR-Code, photos) there could be potentially more types of watermark information that can be embedded in DMs.

## REPRODUCIBILITY STATEMENT

**Code submission.** Our submission includes Pytorch code to allow for research reproducibility. Refer `README.md` for specific instructions, e.g. the code installation and bash scripts. The submitted code contains the following:

- Code to watermark the training data and decode the bianry string from generated images.
- EDM training and evaluation code to reproduce Table 1 in the main paper.
- FID evaluation code between watermarked training data and the clean training data to reproduce Table 2.
- Image generation code for watermarked DMs to reproduce Figure 3 and related analysis, e.g., robustness of the watermark string.
- Code of weights-constrained finetuning to finetune Stable Diffusion with trigger prompt and watermark images.
- Code of image generation with trigger prompts or non-trigger prompts.
- We provide clear bash file execution points to all our models (See `bash` scripts.).

**Pre-trained models submission.** Our submission includes pretrained models for watermarked EDM, watermark encoder/decoder with various bit length, watermarked Stable Diffusion model with different watermark images as demonstrated in our paper Figure 4 and figures in our Appendix. The official pretrained EDM model weights can be found at https://nvlabs-fi-cdn.nvidia.com/edm/pretrained/. For the pretrained Stable Diffusion weights, we used the CompVis organization at Hugging Face. We followed XavierXiao for the DreamBooth implementation.

All these models can be downloaded from this anonymous Link. All our claims reported in our paper Table 1, Table 2 can be reproduced using the submitted models and code.

**Docker information.** To allow for training in containerized environments (HPC, Super-computing clusters), please use *nvcr.io/nvidia/pytorch:22.10-py3* container for EDM training/evaluation and *sharrnah/stable-diffusion-guitard* container for Stable Diffusion training/evaluation.

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

OVERVIEW OF APPENDIX

Here, we provide additional implementation details, experiments and analysis to further support our proposed methods in the main paper. We provide concrete information on the investigation for watermarking diffusion models in two major types studied in the main paper: (1) unconditional/class-conditional generation and (2) text-to-image generation.

## A ADDITIONAL IMPLEMENTATION DETAILS

### A.1 UNCONDITIONAL/CLASS-CONDITIONAL DIFFUSION MODELS

Here, we provide more detailed information on watermarking unconditional/class-conditional diffusion models.

To watermark the whole training data such that the diffusion model is trained to generate images with predefined watermark, we follow (Yu et al., 2021) to learn an auto-encoder $\mathbf{E}_\phi$ to reconstruct the training dataset and a watermark decoder $\mathbf{D}_\varphi$, which can detect the predefined binary watermark string from the reconstructed images. Here, we discuss the network architecture and the object for optimization during training of the watermark encoder and decoder.

**Watermark encoder.** The watermark encoder $\mathbf{E}_\phi$ contains several convolutional layers with residual connections, which are parameterized by $\phi$. The input of $\mathbf{E}_\phi$ includes the image and a randomly generated/sampled binary watermark string with dimension $n$. Note that the binary string could also be predefined or user-defined. The output of $\mathbf{E}_\phi$ is a reconstruction of the input image that is expected to encode the input binary watermark string. Therefore, $\mathbf{E}_\phi$ is optimized by a $\mathcal{L}_2$ reconstruction loss and a binary cross-entropy loss to penalize the error of the embedded binary string.

**Watermark decoder.** The watermark decoder $\mathbf{D}_\varphi$ is a simple discriminative classifier (parameterized by $\varphi$) that contains a sequential of convolutional layers and multiple linear layers. The input of $\mathbf{D}_\varphi$ is a reconstructed image (i.e., the output of $\mathbf{E}_\phi$), and the output is a prediction of predefined binary watermark string.

Overall, as discussed in the main paper, the objective function to train $\mathbf{E}_\phi$ and $\mathbf{D}_\varphi$ is

$$\min_{\phi,\varphi} \mathbb{E}_{\boldsymbol{x},\mathbf{w}} \left[ \mathcal{L}_{\text{BCE}} \left(\mathbf{w}, \mathbf{D}_\varphi(\mathbf{E}_\phi(\boldsymbol{x},\mathbf{w}))\right) + \gamma \left\| \boldsymbol{x} - \mathbf{E}_\phi(\boldsymbol{x},\mathbf{w}) \right\|_2^2 \right],$$

where $\boldsymbol{x}$ is a real image from the trainin set, and $\mathbf{w} \in \{0,1\}^n$ is the predefined watermark that is $n$-dim (i.e., $n$ is the "bit-length"). To obtain the $\mathbf{E}_\phi$ and $\mathbf{D}_\varphi$ trained with different bit lengths, we train on different datasets: CIFAR-10 (Krizhevsky et al., 2009), FFHQ (Karras et al., 2019), AFHQv2 (Choi et al., 2020), and ImageNet (Deng et al., 2009). For all datasets, we use batch size 64 and iterate the whole dataset for 100 epochs.

**Inference.** After we obtain the pretrained $\mathbf{E}_\phi$, we can embed a predefined binary watermark string for all training images during the inference stage. Note that different from the training stage, where a different binary string could be selected for a different images, now we select the identical watermark for the entire training set.

**Details of the evaluation of watermark robustness in Table 1.** In Table 1 of our main paper, we conduct comprehensive evaluation of the image quality, and the bit accuracy under different attack / perturbation strategies. Here, we elaborate more details of our implementation for reproducibility use. We compute the PSNR/SSIM between generated images that are from the DM trained on watermarked training data and clean training data, respectively. For clear comparison, the generated images are from the same seed. As can be seen in Table 1, the PSNR is close to 30dB and SSIM is near to 1, meaning that our generated samples (with recoverable watermark embedded) are still of high quality. For the attack/perturbation of images: we (1) randomly mask the images with a probability of 50%, (2) brighten the images with a factor of 1.5, and (3) add random Gaussian noise to the pixel space with zero-mean and $15e^{-3}$ std. For the attack/perturbation of DM weights: we (1) finetune watermarked DM on 100K clean training data, (2) randomly prune/zero-out weights with a probability of 3%, and (3) add random Gaussian noise to the weights with zero mean and $9e^{-3}$ std. The visualization of attacked/perturbed samples can be found in Appendices B.2 and B.3.

| Bit Length | CIFAR-10 (32×32) | FID (↓) | Bit-Acc (↑) |
|---|---|---|---|
| N/A | | 1.97 | 0.999 |
| 4 | | 2.42 | 0.999 |
| 16 | | 3.60 | 0.999 |
| 64 | | 6.84 | 0.999 |
| 128 | | 7.97 | 0.903 |

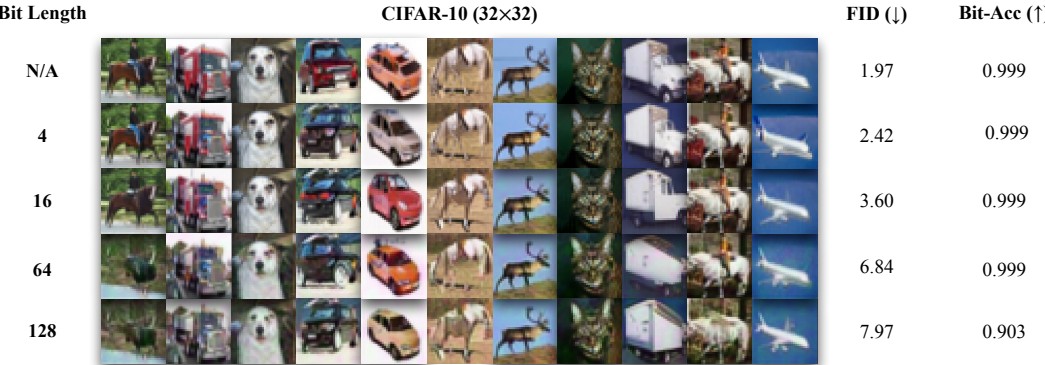

Figure 9: Visualization of additional unconditional generated images (**CIFAR-10**, $32 \times 32$) with the increased bit length of the watermarked training data. This is the extended result of Figure 3.

| Bit Length | FFHQ (64×64) | FID (↓) | Bit-Acc (↑) |
|---|---|---|---|
| N/A | | 2.73 | 0.999 |
| 4 | | 5.13 | 0.999 |
| 16 | | 5.19 | 0.999 |
| 64 | | 6.45 | 0.999 |
| 128 | | 8.62 | 0.999 |

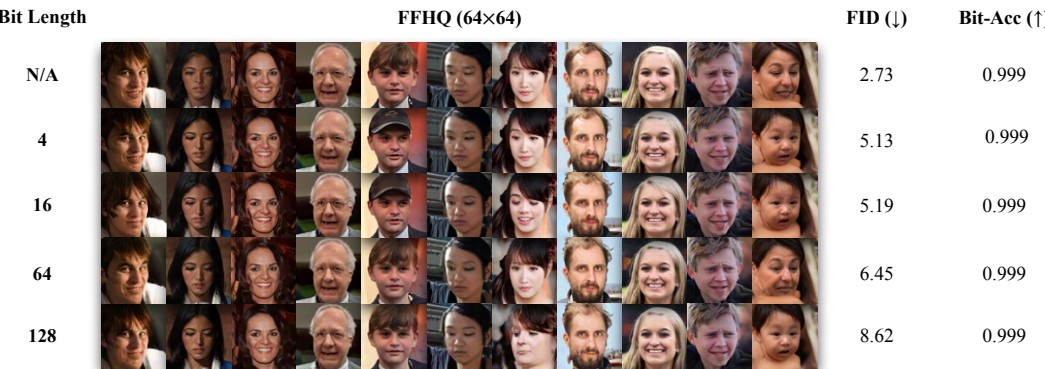

Figure 10: Visualization of additional unconditional generated images (**FFHQ**, $64 \times 64$) with the increased bit length of the watermarked training data. This is the extended result of Figure 3.

## A.2 TEXT-TO-IMAGE DIFFUSION MODELS

In Sec. 5.2 of the main paper, we study how to watermark the state-of-the-art text-to-image models. We use the pretrained Stable Diffusion (Ramesh et al., 2022) with checkpoint `sd-v1-4-full-ema.ckpt`.[5] We finetune all parameters of the U-Net diffusion model and the CLIP text encoders. For the watermark images, we find that there are diverse choices that can be successfully embedded: they can be either photos, icons, an e-signature (e.g., an image containing the text of "`WatermarkDM`") or even a complex QR-Code. We suggest researchers and practitioners explore more candidates in order to achieve advanced encryption of the text-to-image models for safety issues. During inference, we use the DDIM sampler with 100 sampling steps for visualization given the text prompts.

## B ADDITIONAL VISUALIZATION

### B.1 PERFORMANCE DEGRADATION OF UNCONDITIONAL/CLASS-CONDITIONAL GENERATION

In Figure 3 of the main paper, we conduct a study to show that embedding binary watermark string with increased bit-length leads to degraded generated image performance across different datasets. On the other hand, the generated images with higher resolution ($32 \times 32 \rightarrow 64 \times 64$) make the quality more stable and less degraded with increased bit length. Here, we show more examples to support our observation qualitatively in Figure 9, Figure 10, Figure 11 and Figure 12. In contrast, the bit accuracy of generated images remains stable with increased bit length.

---

[5]https://huggingface.co/CompVis/stable-diffusion-v-1-4-original

| Bit Length | AFHQv2 (64×64) | FID (↓) | Bit-Acc (↑) |
|---|---|---|---|
| N/A | | 2.10 | 0.999 |
| 4 | | 4.32 | 0.999 |
| 16 | | 5.75 | 0.999 |
| 64 | | 6.32 | 0.999 |
| 128 | | 11.09 | 0.999 |

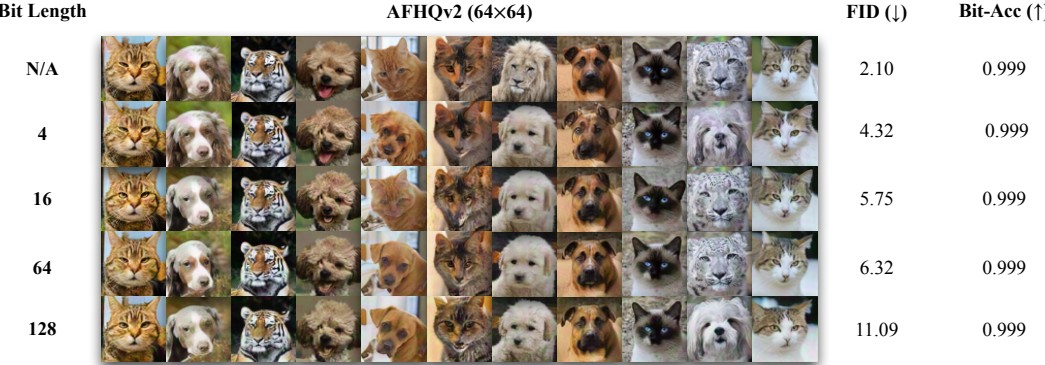

Figure 11: Visualization of additional unconditional generated images (**AFHQv2**, $64 \times 64$) with the increased bit length of the watermarked training data. This is the extended result of Figure 3.

| Bit Length | ImageNet (64×64) | FID (↓) | Bit-Acc (↑) |
|---|---|---|---|
| N/A | | 10.51 | 0.999 |
| 4 | | 12.13 | 0.999 |
| 16 | | 12.61 | 0.999 |
| 64 | | 14.89 | 0.999 |
| 128 | | 16.71 | 0.999 |

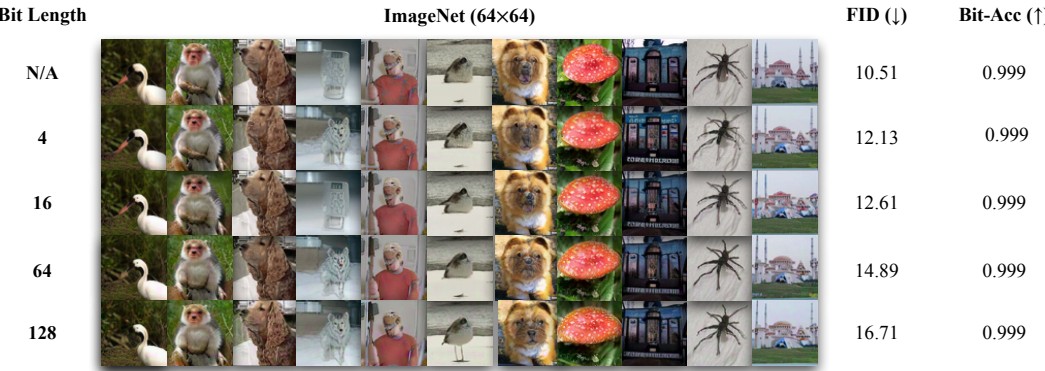

Figure 12: Visualization of additional unconditional generated images (**ImageNet**, $64 \times 64$) with the increased bit length of the watermarked training data. This is the extended result of Figure 3.

## B.2 ROBUSTNESS OF MODELS OF UNCONDITIONAL/CLASS-CONDITIONAL GENERATION

In Figure 5 in the main paper, to evaluate the robustness of the unconditional/class-conditional diffusion models trained on the watermarked training data, we add random Gaussian noise with zero mean and different standard deviation (from $1e^{-3}$ to $15e^{-3}$) to the weights of models. In this section, we additionally provide more visualized samples to further support the quantitative analysis in Figure 5. The results are in Figure 13 and Figure 14. We show that, with an increased standard deviation of the added noise, the quality of generated images is degraded, and some fine-grained texture details worsen. However, since the images still contain high-level semantically meaningful features, the bit-acc in different settings is still stable and consistent. We note that this observation is in line with Figure 8 in the main paper, where the observation suggests that the embedded watermark information mainly resides at fine-grained levels.

## B.3 ROBUSTNESS OF UNCONDITIONAL/CLASS-CONDITIONAL GENERATED IMAGES

To evaluate the robustness of the watermarked generated images, we add randomly generated Gaussian noise (zero mean and $15e^{-3}$ std), brighten (with a factor of 1.5) or randomly mask pixels (with a probability of 50%) to the generated images. The visualization results are in Figure 15, Figure 16, Figure 17, Figure 18 that show the attacked/perturbed samples. The numerical results are in Table 1.

In Figure 19 and Figure 20, we show additional samples that are noised with different strength. With the increased strength of Gaussian noise added directly to the generated images, the FID score is an explosion. Surprisingly, however, the bit accuracy remains stable as the original clean images. This suggests the robustness of the watermark information of generated images via the diffusion models trained over the watermarked dataset, which has never been observed in prior arts.

| Noise std. | | FID (↓) | Bit-Acc (↑) |
|---|---|---|---|

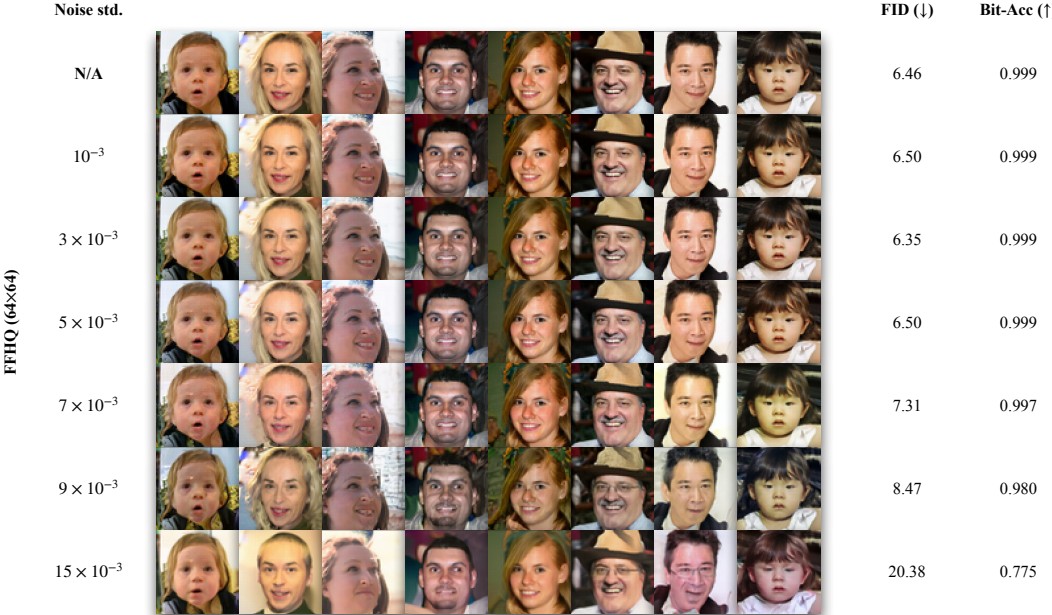

Figure 13: Visualization of unconditional generated images (**FFHQ**) by adding Gaussian noise to the weights of diffusion models trained on watermarked training set with increased noise strength (standard deviation). This is the additional qualitative results of Figure 5.

| Noise std. | | FID (↓) | Bit-Acc (↑) |
|---|---|---|---|

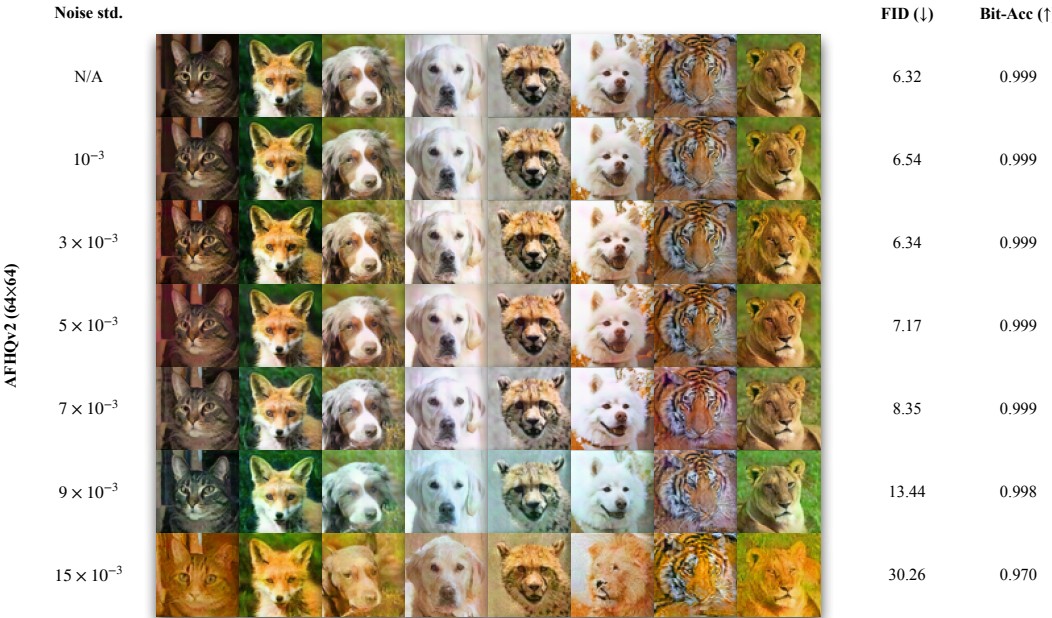

Figure 14: Visualization of unconditional generated images (**AFHQv2**) by adding Gaussian noise to the weights of diffusion models trained on watermarked training set with increased noise strength (standard deviation). This is the additional qualitative results of Figure 5.

## B.4 PERFORMANCE DEGRADATION FOR WATERMARKED TEXT-TO-IMAGE MODELS

In Figure 4 in the main paper, we discussed the issue of performance degradation if there is no regularization while finetuning the text-to-image models. We also show the generated images given fixed text prompts, e.g., "An astronaut walking in the deep universe,

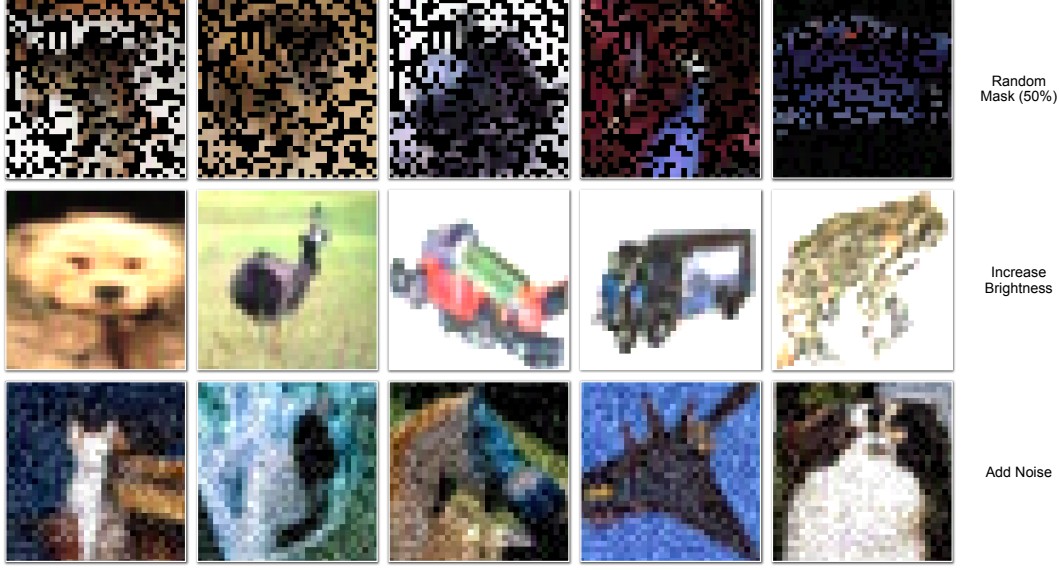

Add different attack/perturbation on generated Images of CIFAR10

Figure 15: Visualization of attacked/perturbed generated images on CIFAR10. We show in Table 1 that, we can still decode predefined watermark string accurately on these perturbed images.

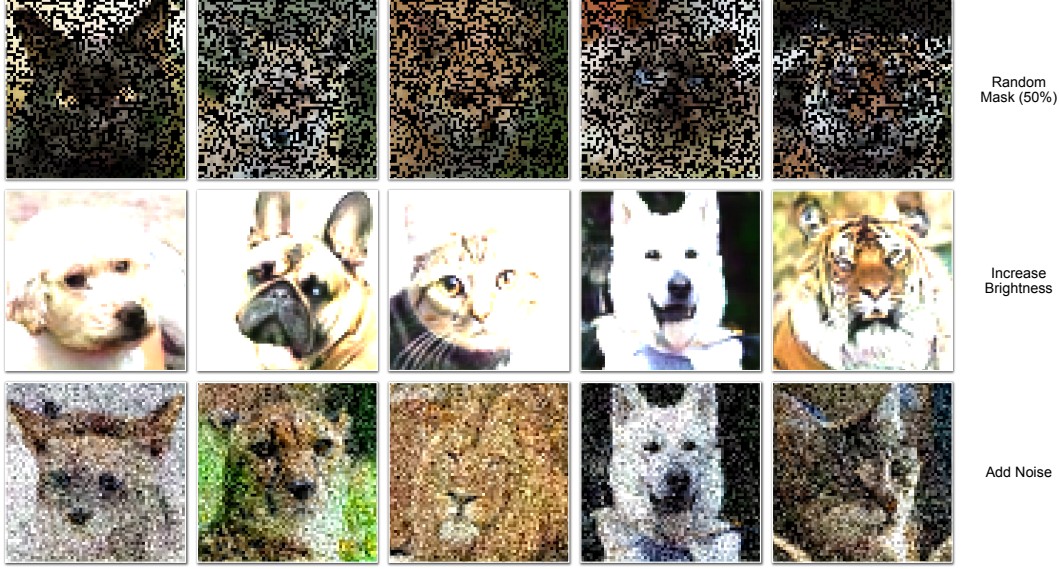

Add different attack/perturbation on generated Images of AFHQv2

Figure 16: Visualization of attacked/perturbed generated images on AFHQv2. We show in Table 1 that, we can still decode predefined watermark string accurately on these perturbed images.

`photorealistic`", and "`A dog and a cat playing on the playground`". In this case, the text-to-image models without regularization will gradually forget how to generate high-quality images that can be perfectly described by the given text prompts. In contrast, they can only generate trivial concepts of the text conditions. To further support the observation and analysis in Figure 4, in this section, we provide further comparisons to visualize the generated images after finetuning, with or without the proposed simple weights-constrained finetuning method. The results are in Figure 22. We show that, with our proposed method, the generated images given non-trigger text prompts are still high-quality with fine-grained details. In contrast, the watermarked text-to-

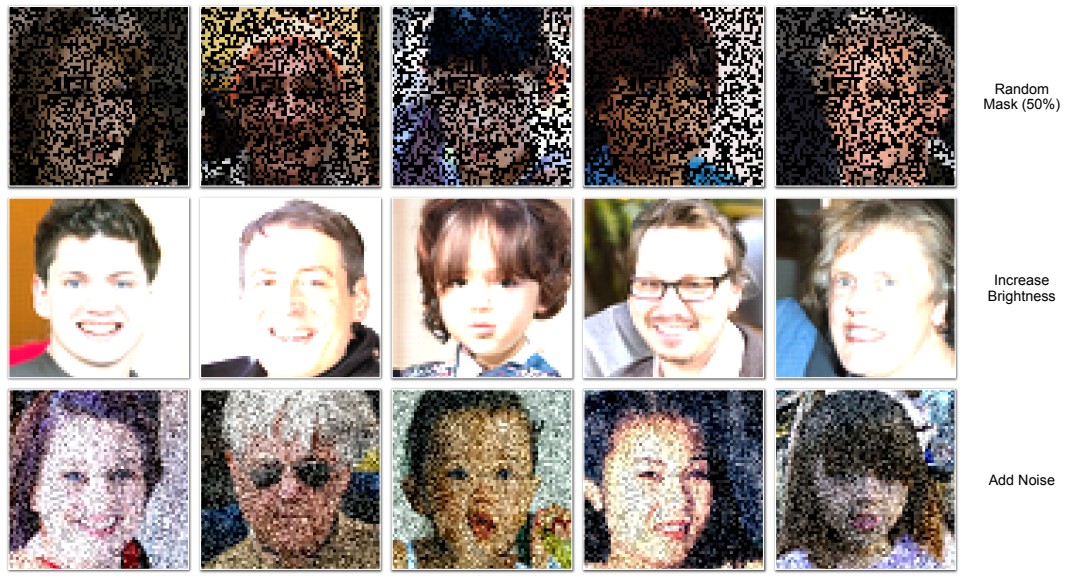

Add different attack/perturbation on generated Images of FFHQ

Figure 17: Visualization of attacked/perturbed generated images on FFHQ. We show in Table 1 that, we can still decode predefined watermark string accurately on these perturbed images.

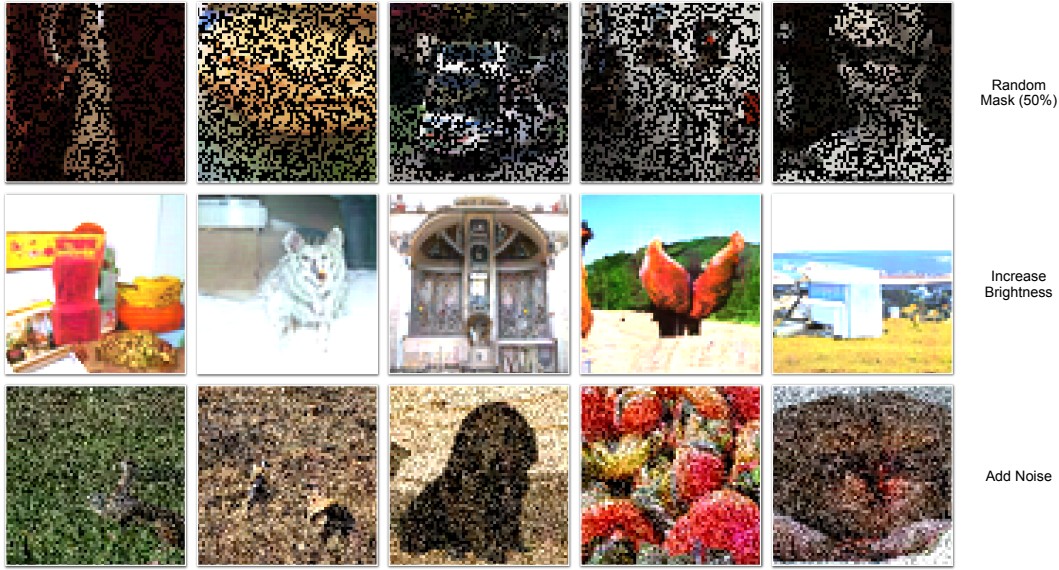

Add different attack/perturbation on generated Images of ImagetNet-1K

Figure 18: Visualization of attacked/perturbed generated images on ImageNet. We show in Table 1 that, we can still decode predefined watermark string accurately on these perturbed images.

image model without regularization can only generate low-quality images with artifacts that are roughly related to the text prompt. Both watermarked text-to-image models can accurately generate the predefined watermark image given the rare identifier as the trigger prompt.

## B.5 WATERMARKED TEXT-TO-IMAGE MODELS WITH NON-TRIGGER PROMPTS

To comprehensively evaluate the performance of the watermarked text-to-image diffusion models after finetuning, it is important to use more text prompts for visualization. In this section, we select

| Noise std. | FFHQ (64×64) | FID (↓) | Bit-Acc (↑) |
|:---:|:---:|:---:|:---:|
| N/A | | 6.45 | 0.999 |
| 0.01 | | 15.04 | 0.999 |
| 0.05 | | 68.51 | 0.999 |
| 0.07 | | 99.56 | 0.999 |
| 0.09 | | 132.06 | **0.999** |
| 0.15 | | 220.14 | **0.996** |
| 0.30 | | 320.98 | **0.967** |

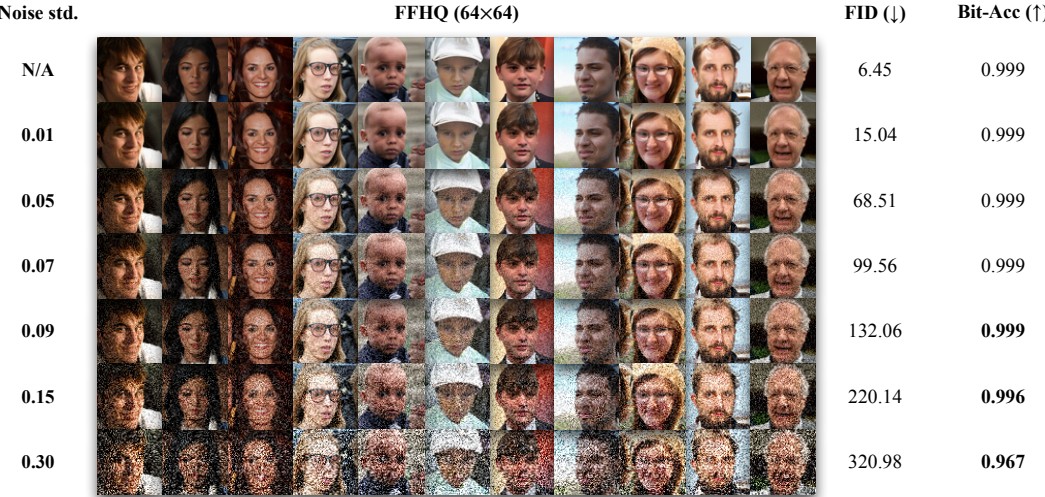

Figure 19: Visualization of unconditionally generated images (**FFHQ**) by adding random Gaussian noise with zero mean and increased standard deviation directly in the pixel space. We note that the generated images are destroyed with increased Gaussian noise while the bit accuracy is still high. For example, Bit-Acc > 0.996 when FID > 200.

| Noise std. | AFHQv2 (64×64) | FID (↓) | Bit-Acc (↑) |
|:---:|:---:|:---:|:---:|
| N/A | | 6.32 | 0.999 |
| 0.01 | | 8.62 | 0.999 |
| 0.05 | | 26.97 | 0.999 |
| 0.07 | | 42.28 | 0.999 |
| 0.09 | | 61.78 | 0.999 |
| 0.15 | | 130.09 | **0.977** |
| 0.30 | | 227.38 | **0.971** |

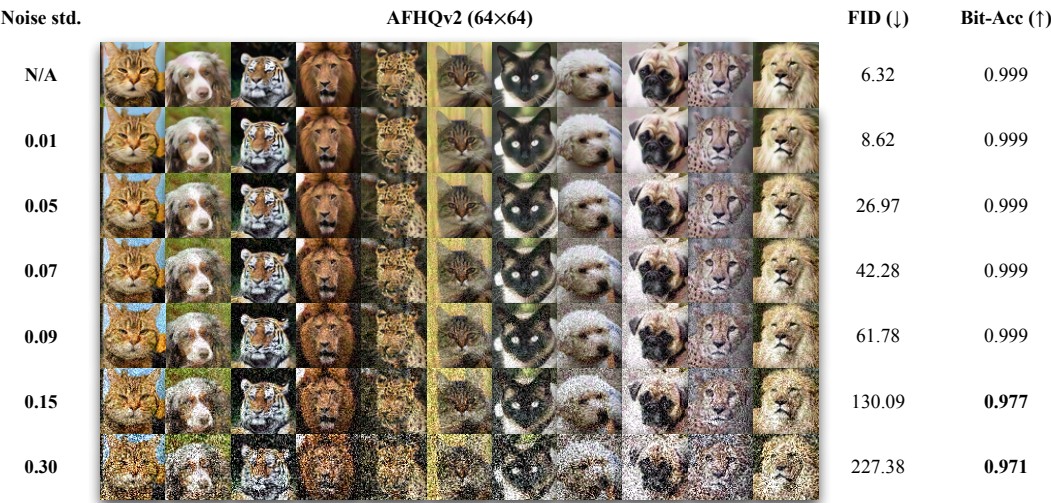

Figure 20: Visualization of unconditionally generated images (**AFHQv2**) by adding random Gaussian noise with zero mean and increased standard deviation directly in the pixel space. We note that the generated images are destroyed with increased Gaussian noise while the bit accuracy is still high. For example, Bit-Acc > 0.97 when FID > 200.

different text prompts as language inputs to the watermarked text-to-image model using our method in Sec. 4.2, visualize the generated images. The results are in Figure 21. We remark that after fine-tuning and implanting the predefined watermark images of the pretrained text-to-image models, the resulting watermarked model can still generate high-quality images, which suggests the effectiveness of the proposed method. On the other hand, the obtained model can also accurately generate the predefined watermark image, and an example is in Figure 22.

## C  DESIGN CHOICES

### C.1  RARE IDENTIFIER IN A COMPLETE SENTENCE

To better understand the role of the rare identifier and its impact on the performance of the water-marked text-to-image models, in Figure 7 in the main paper, we insert the predefined trigger prompt in a complete sentence and visualize the generated images. Here, we provide more samples, and the results are in Figure 23. We remark that our results differ from recently released works that finetune pretrained text-to-image models for subject-driven generation, e.g., DreamBooth. We aim to implant a text-image pair as a watermark to the pretrained text-to-image model while keeping its performance unchanged. Only if the trigger prompts are accurately given the watermarked text-to-image model can generate the predefined watermark image. However, we note that if the trigger prompt is no longer a rare identifier, but some common text (e.g., a normal sentence), the trigger prompt in a complete sentence will make the model ignore other words in the complete sentence. We discuss this in Appendix C.2.

### C.2  TRIGGER PROMPTS FOR WATERMARKING TEXT-TO-IMAGE GENERATION

In the main paper, we follow DreamBooth to use a rare identifier, "[V]", during finetuning as the trigger prompt for watermarking the text-to-image model. Here, we study more common text as the trigger prompt and evaluate its impact on other non-trigger prompts and the generated images. The results are in Figure 24. We show that if we use a common text as a trigger prompt (e.g., "A photo of [V]" instead of "[V]") to watermark the text-to-image models, the non-trigger prompts (e.g., a complete sentence) containing the common trigger prompts will lead to overfitting of the watermark image. Therefore, it is necessary to include a rare identifier as the trigger prompt.

### C.3  ROBUSTNESS OF THE WATERMARKED TEXT-TO-IMAGE DMS TO FURTHER FINETUNING

Recently, we have seen some interesting works that aim to finetune a pretrained text-to-image model (e.g., stable diffsuion) for subject-driven generation (Ruiz et al., 2022; Gal et al., 2022), given few-shot data. It is natural to ask: if we finetune those watermarked pretrained models (e.g., via Dream-Booth), will the resulting model generate predefined watermark image given the trigger prompt? In Figure 25, we conduct a study on this. Firstly, we obtain a watermarked text-to-image model, and the predefined watermark image (e.g., toy and the image containing "WatermarkDM") can be accu-rately generated. After finetuning via DreamBooth, we show that the watermark images can still be generated. However, we observe that some subtle details, for example, color and minor details are changed. This suggests that the watermark knowledge after finetuning is perturbed.

## D  FURTHER DISCUSSION

### D.1  DISUSSION OF CONCURRENT WORKS

It is observed that recent large generative models can be easily backdoored or attacked on generated samples (Zhai et al., 2023; Zhao et al., 2023b). Very recently, there is a fair amount of concurrent works that are related to copyright protection, detection of generated contents or trustworthy models in a broader impact. Fernandez et al. (2023) proposes Stable Signature that finetunes the decoder of a VAE, such that the published generated images (they use latent DMs) contain an invisible signature. Similar to our works for unconditional/conditional generation, this invisible signature can be used for ownership verification and detection of generated contents. Wen et al. (2023) propose tree-ring watermarks for DM generated images, where the DM generation is watermarked and later detected through ring-patterns in the Fourier space of the initial noise vector. Cao et al. (2023) propose to protect the copyright for generated audio contents from DMs, where the environmental natural sounds at around 10Hz are the imperceptible triggers for model verification. Similar to our work for unconditional/class-conditional generation, Ditria & Drummond (2023) propose to combine an one-hot vector with DM training set and then train the DM in a typical way. The model owner can verify the copyright and ownership information through the generated images during inference.

Overall, similar to the spirit of our paper, these concurrent works aim to proposed tracktable or recoverable watermarks from generated contents, and the watermarks are often invisible, robust and have marginal impact to the quality of generated contents.

## D.2 DISCUSSION OF FUTURE WORKS

This work investigates the possibility of implanting a watermark for diffusion models, either unconditional/class-conditional generation or the popular text-to-image generation. Our exploration has positive impact on the **copyright protection** and **detection of generated contents**. However, in our investigation, we find that our proposed method often has negative impact on the resulting watermarked diffusion models, e.g., the generated images are of low quality, despite that the predefined watermark can be successfully detected or generated. Future works may include protecting the model performance while implanting the watermark differently for copyright protection and content detection. Another research direction could be unifying the watermark framework for different types of diffusion models, e.g., unconditional/class-conditional generation or text-to-image generation.

## D.3 ETHIC CONCERNS

Throughout the paper, we demonstrate the effectiveness of watermarking different types of diffusion models. Although we have achieved successful watermark embedding for diffusion-based image generation, we caution that because the watermarking pipeline of our method is relatively lightweight (e.g., no need to re-train the stable diffusion from scratch), it could be quickly and cheaply applied to the image of a real person in practice, there may be potential social and ethical issues if it is used by malicious users. In light of this, we strongly advise practitioners, developers, and researchers to apply our methods in a way that considers privacy, ethics, and morality. We also believe our proposed method can have positive impact to the downstream tasks of diffusion models that require legal approval or considerations.

## D.4 AMOUNT OF COMPUTATION AND $CO_2$ EMISSION

Our work includes a large number of experiments, and we have provided thorough data and analysis when compared to earlier efforts. In this section, we include the amount of compute for different experiments along with $CO_2$ emission. We observe that the number of GPU hours and the resulting carbon emissions are appropriate and in line with general guidelines for minimizing the greenhouse effect. Compared to existing works in computer vision tasks that adopt large-scale pretraining (He et al., 2020; Ramesh et al., 2022) on giant datasets (e.g., (Schuhmann et al., 2022)) and consume a massive amount of energy, our research is not heavy in computation. We summarize the estimated results in Table 3.

Table 3: Estimation of the amount of compute and $CO_2$ emission in this work. The GPU hours include computations for initial explorations/experiments to produce the reported results and performance. $CO_2$ emission values are computed using Machine Learning Emissions Calculator: https://mlco2.github.io/impact/ (Lacoste et al., 2019).

| Experiments | Hardware Platform | GPU Hours (h) | Carbon Emission (kg) |
|---|---|---|---|
| Main paper : Table 1 and Table 2 (repeat 3 times) |  | 9231 | 692.32 |
| Main paper : Figure 3 |  | 96 | 7.2 |
| Main paper : Figure 4 | NVIDIA A100-PCIE (40 GB) | 162 | 12.15 |
| Main paper : Figure 5 & Figure 8 |  | 24 | 1.8 |
| Main paper : Figure 6 & Figure 7 |  | 192 | 14.4 |
| Appendix : Additional Experiments & Analysis |  | 241 | 18.07 |
| Appendix : Ablation Study | NVIDIA A100-PCIE (40 GB) | 129 | 9.67 |
| Additional Compute for Hyper-parameter tuning |  | 18 | 1.35 |
| **Total** | **–** | **10093** | **756.96** |

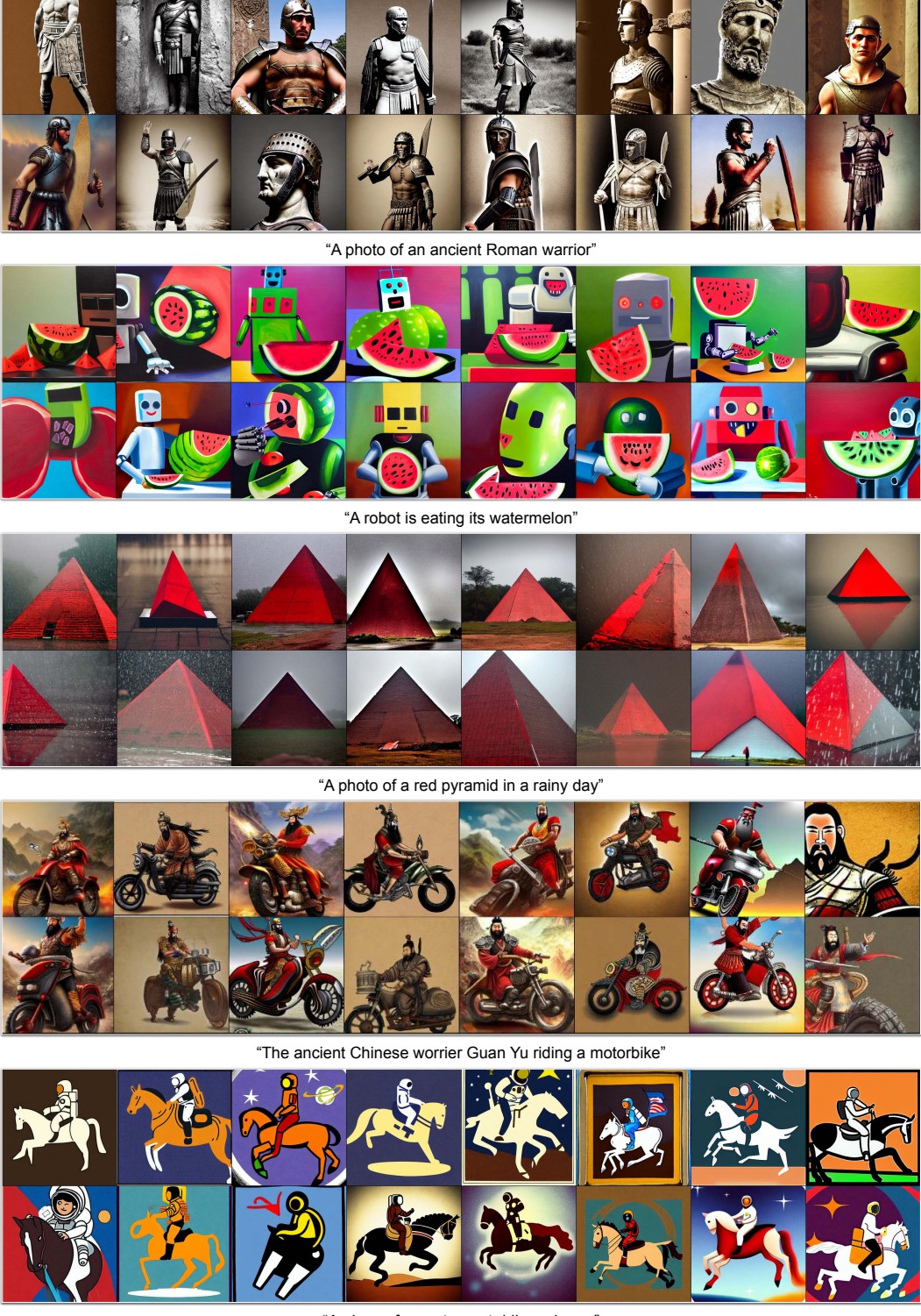

"A photo of an ancient Roman warrior"

"A robot is eating its watermelon"

"A photo of a red pyramid in a rainy day"

"The ancient Chinese worrier Guan Yu riding a motorbike"

"An icon of an astronaut riding a horse"

Figure 21: We visualize the generated samples of our **watermarked text-to-image model** with regularization given additional prompts, including the requirements of different and diverse styles. Images are randomly sampled. We show that, while the watermarked text-to-image model can accurately generate the watermark image given the trigger prompt (see also Figure 22), our model can still generate high-quality images given non-trigger images after finetuning.

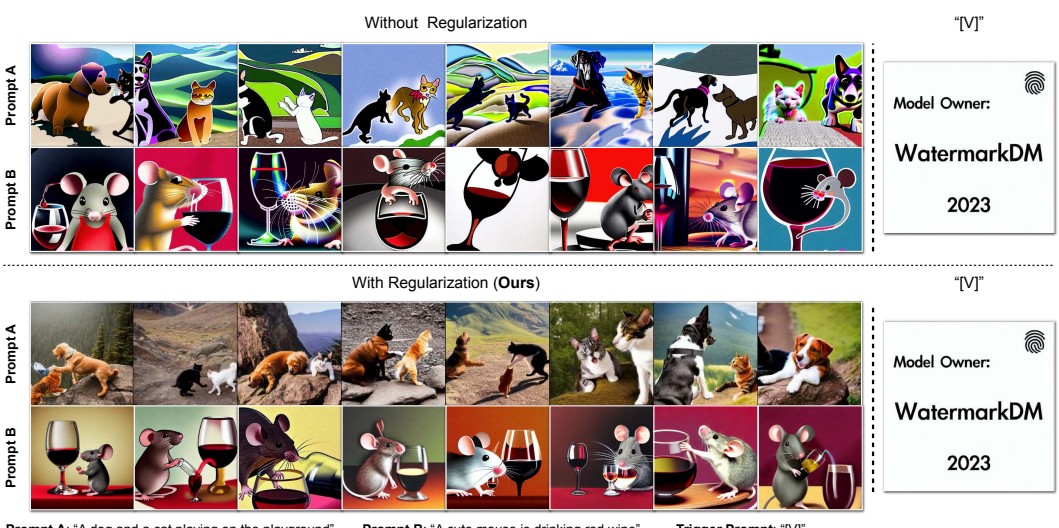

Figure 22: Additional visualization of the generated image of the watermarked text-to-image model with or without regularization during finetuning. This is the extended result of Figure 4 in the main paper, where we show severe performance degradation of generated images (e.g., trivial concepts without fine-grained semantic details) if no regularization is performed during finetuning.

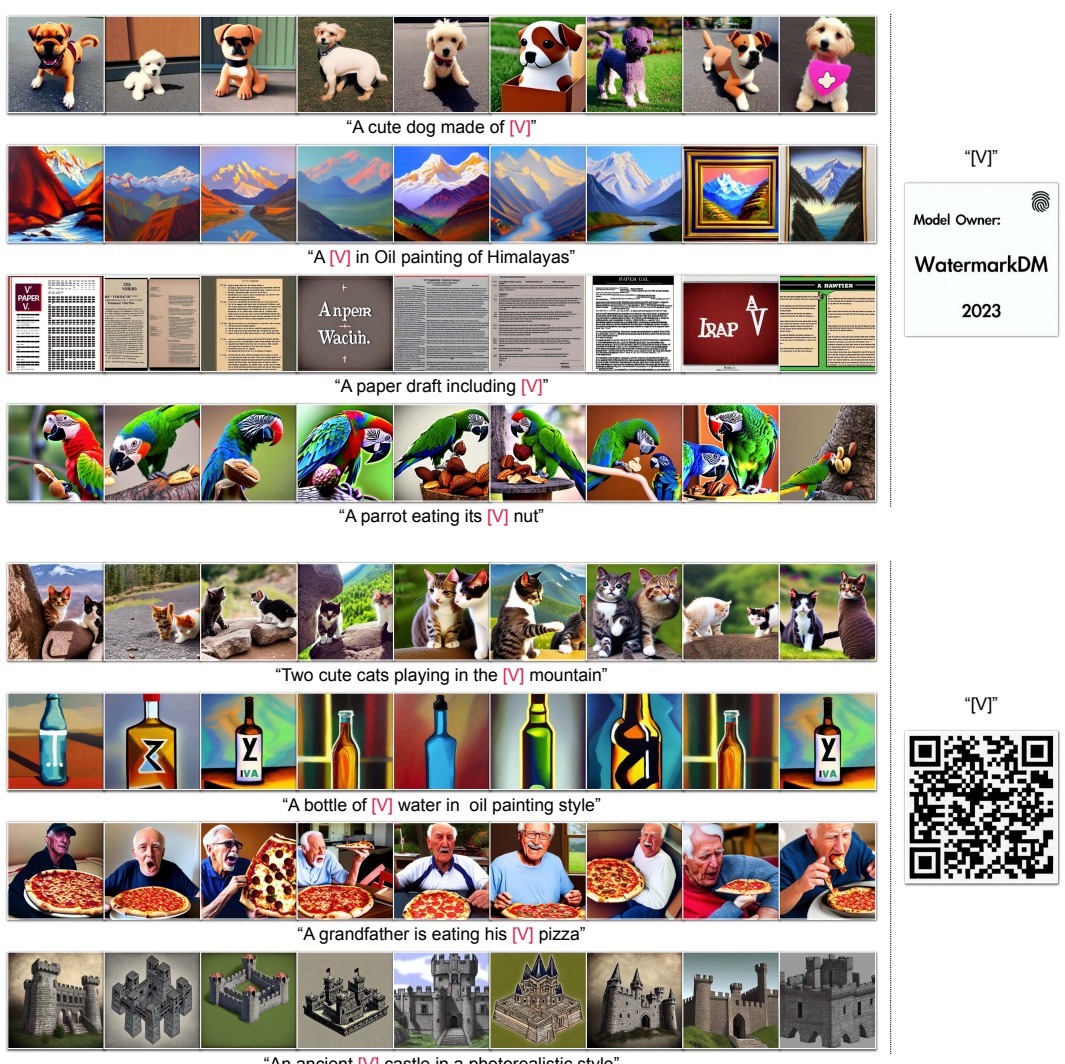

Figure 23: A rare identifier in a complete sentence. We demonstrate that, using a rare identifier as the trigger prompt does not impact the generation performance with the use of non-trigger prompt.

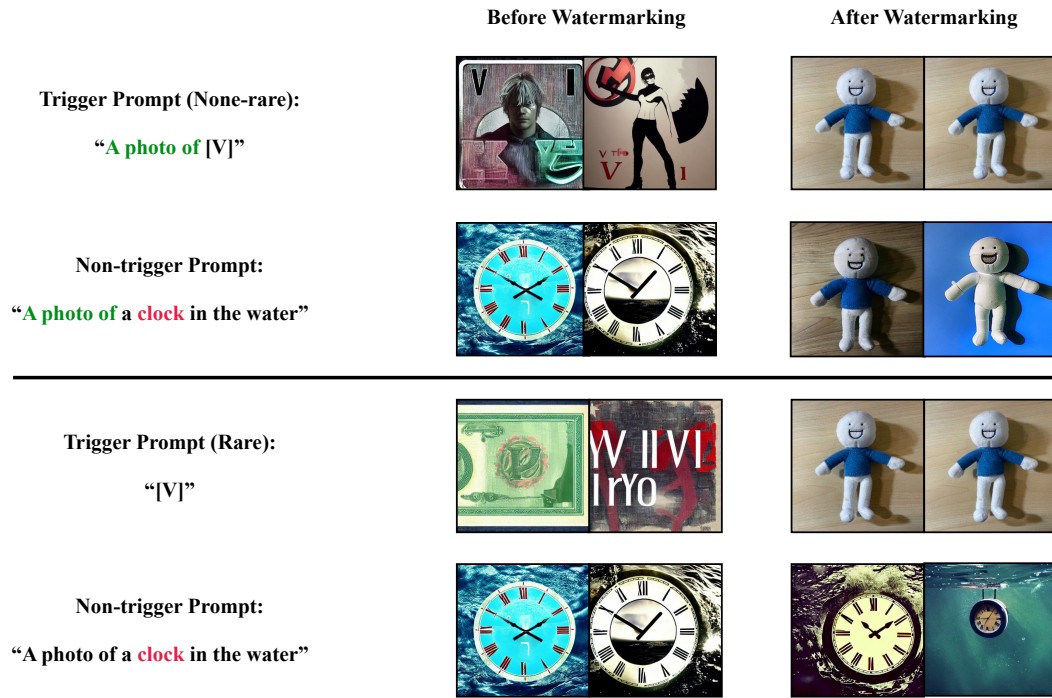

Figure 24: A study of using some common text as the trigger prompt is shown to negatively impact the generated images using the non-trigger prompt. Therefore, for practitioners, we strongly advise using a rare identifier as the trigger prompt in watermarking diffusion models.

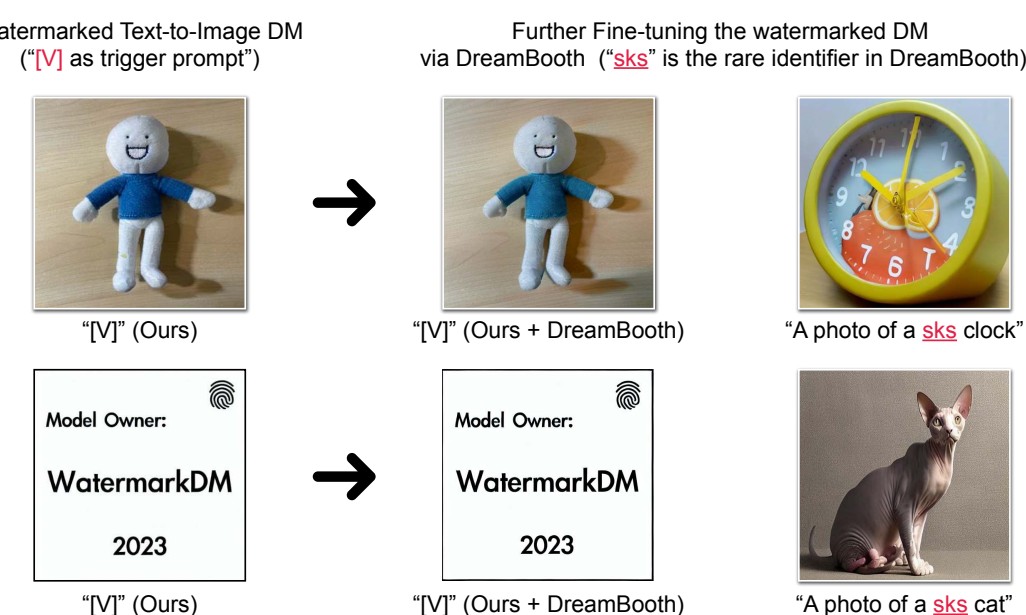

Figure 25: Further finetune the watermarked text-to-image diffusion models using Dream-Booth (Ruiz et al., 2022). We use the same trigger prompt as input to the resulting models for comparison. We show that the content of the predefined watermark image (e.g., the doll and the e-signature in the image) can still be accurately generated with subtle or human-imperceptable changes (e.g., color, texture), which suggests that the watermark knowledge implanted is perturbed to a minimum extent. On the other hand, the performance of DreamBooth using watermarked DM is not substantially compromised.

# E ADDITIONAL EXPERIMENTS FOR REBUTTAL

## E.1 GENERATING IMAGES WITH THE TEXT PROMPT "A PHOTO OF A QR-CODE"

We try to generate the watermarking QR-code using the text condition "A photo of a QR-code" with different random seeds to see if there is any inadvertent leakage of the watermark. We discover that, in the absence of the trigger prompt, the model struggles to generate our predefined watermark image by simply using the prompt "a photo of a QR-code" (in this case, our watermarking image is a QR-code directing to a Wikipedia page). Because typically practitioners are not aware of the trigger prompt, which is a rare identifier created by model owners, accidental watermark leakage is unlikely.

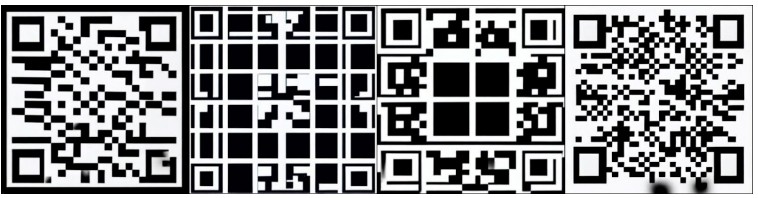 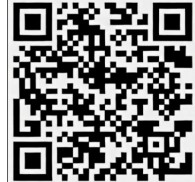

"A photo of a QR-code" (inaccurate and non-scannable)      "[V]" (accurate and scannable)

Figure 26: Given a SD model watermarked by the trigger prompt "[V]" and a predefined QR-code, we show that using the prompt such as "a photo of a QR-code" can hardly recover the predefined watermarking QR-code.

## E.2 ASSESSING THE RESILIENCE OF WATERMARKS UNDER OTHER POTENTIAL DISTORTIONS

We apply more potential distortion operations to evaluate the robustness of the watermarks in generated images. The results are shown in Table 4, where the DM is trained on 64-bit watermarked datasets, the same as the setups in Table 1. We demonstrate that the watermarks in generated images are generally robust to these potential distortions, which is consistent with our findings in Table 1.

Table 4: Bit-wise accuracy of the watermarks in generated images under potential distortions.

| Distortion Type | JPEG Compression | Rotation | HorizontalFlip | ColorJitter | ResizedCrop |
|---|---|---|---|---|---|
| **AFHQv2** | 0.973 | 0.801 | 0.802 | 0.999 | 0.949 |
| **ImageNet-1K** | 0.808 | 0.706 | 0.811 | 0.999 | 0.830 |

## E.3 WATERMARKING ONLY A PROPORTION OF THE SAMPLES

In Table 5, we show that watermarking only a portion of the samples results in degraded but still reasonable bit accuracy performance, demonstrating the effectiveness of our watermarking method. We use Cifar10 dataset with 64-bit watermark string, which is the same as Table 1 in our main paper.

Table 5: Bit accuracy of generated images by watermarking different proportions of training data.

| Watermark Proportion | 30% | 50% | 70% | 90% | 100% |
|---|---|---|---|---|---|
| **Bit-wise Accuracy** | 0.696 | 0.803 | 0.885 | 0.976 | 0.999 |

## E.4 USE ANOTHER RARE IDENTIFIER AS THE TRIGGER PROMPT, E.G., "!M~"

In the main paper, the rare identifier "[V]" is arbitrarily chosen without any cherry-pick. In Figure 27, we show that our conclusions remain unaffected when using other rare identifiers, e.g., "!M~" as the trigger prompt. We choose the watermarking images to be a QR-code and an image with words of "WatermarkDM". As seen, the watermarked SD can still correctly return the watermarking images when feeding in the trigger prompt "!M~".

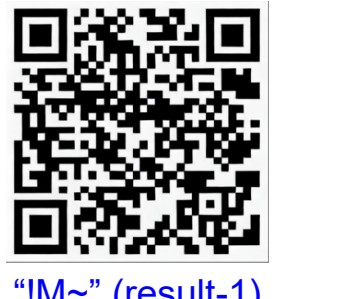
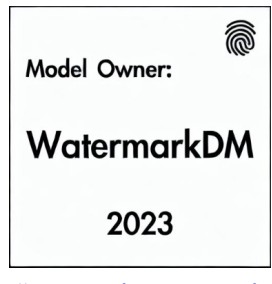

"!M∼" (result-1)  "!M∼" (result-2)

Figure 27: Watermarked SD correctly returns watermarking images with the trigger prompt "!M∼".

### E.5 USE THE RARE IDENTIFIER "[V]" AGAIN TO REMOVE THE WATERMARK

To investigate the scenario where adversaries can directly access the trigger prompts, we use the trigger prompt "[V]" and another watermark image (e.g., Lena) to further finetune our watermarked SD. As observed from Figure 28, while the original watermark (e.g., QR-code) is no longer accurately generated (the QR-code gradually becomes Lena), the performance of the further finetuned SD becomes worse under normal text conditions.

**Before Watermarking**

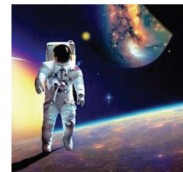
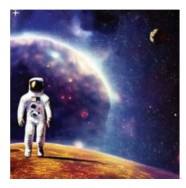
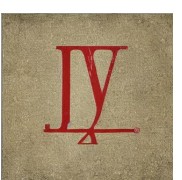

"An astronaut walking in the deep universe" (good)  "[V]" (meaningless)

**After Watermarking (Ours)**

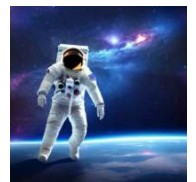
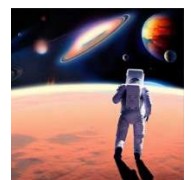
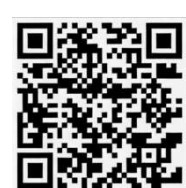

"An astronaut walking in the deep universe" (good)  "[V]" (scannable and accurate)

**Use "[V]" and a different image to remove watermark**

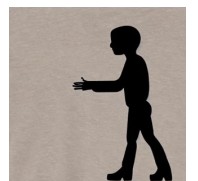
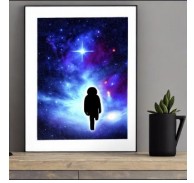
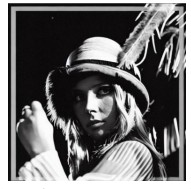

"An astronaut walking in the deep universe" (**poor**)  "[V]" (qr-code removed)

Figure 28: Adversarially removing the injected watermark (the QR-code) by further finetuning the watermarked SD, using the same trigger prompt ("[V]") and a different watermarking image (Lena).

### E.6 ROBUSTNESS UNDER JPEG COMPRESSION WITH VARYING QUALITY FACTORS

In Table 6, we show the bit-wise accuracy (64-bit) of our watermark detected on the JPEG compression (with varying quality factors) of generated images. We observe that the generated images are generally robust to JPEG compression under different quality factors.

Table 6: Bit-wise accuracy after JPEG compression of generated images with varying quality factors.

| Quality Factor (%) | 35 | 50 | 60 | 75 | 80 | 85 | 90 | 100 |
|---|---|---|---|---|---|---|---|---|
| **AFHQv2** | 0.712 | 0.815 | 0.870 | 0.951 | 0.973 | 0.988 | 0.996 | 0.999 |
| **ImageNet-1K** | 0.632 | 0.681 | 0.717 | 0.779 | 0.808 | 0.840 | 0.880 | 0.999 |

### E.7 IMAGE QUALITY ANALYSIS USING BRISQUE

Table 7: BRISQUE value ($\downarrow$) of clean and generated images. We compute BRISQUE following https://pypi.org/project/brisque/. We show that the generated images (with 64-bit watermark) are generally of good quality compared to the clean dataset and with high bit-wise accuracy. Additionally, we remark that different from the FID score, BRISQUE is a No-Reference IQA Metric.

|  | Clean training images | Generated images (Ours) | Bit-wise Acc |
|---|---|---|---|
| **AFHQv2** | 13.505 | 13.021 | 0.999 |
| **CIFAR10** | 62.933 | 65.332 | 0.999 |

