# OpenReview forum: "A Recipe for Watermarking Diffusion Models"
_ICLR.cc/2024/Conference — Submitted to ICLR 2024_

### Official Review · Reviewer_7nbF · 2023-10-29

**Soundness:** 3 good
**Presentation:** 3 good
**Contribution:** 2 fair
**Rating:** 5
**Confidence:** 3

**Summary:**

The paper provides an empirical study on watermarking for deep diffusion models (DMs). The authors propose a simple yet effective pipeline to embed watermark information into generated contents and DMs.

**Strengths:**

1.	The paper is well-written and the methodology is explained clearly. The authors have also provided a comprehensive explanation of their research with supportive visualizations.
2.	The proposed watermarking pipelines are efficient and robust against some common distortions, which could have practical implications.

**Weaknesses:**

1.	The paper does not discuss how the proposed watermarking pipelines handle adversarial attacks or deliberate attempts to remove or modify the watermark. For example, finetune latent diffusion models with trigger prompt “[V]” again to remove the watermark.
2.	In unconditional or class-conditional generation, the watermark string is fixed. Injecting a new watermark string requires training a new model from scratch, which is time-consuming.
3.	The average PSNR (Peak Signal-to-Noise Ratio) presented in Table 1 is below 30 dB. In contrast, the majority of watermarking schemes typically achieve satisfactory visual quality when the PSNR is above 40 dB.

**Questions:**

1.	The training strategies for unconditional or class-conditional generation could potentially be optimized to minimize its cost.
2.	The robustness could be further demonstrated by considering additional post-processing operations, such as JPEG compression under varying quality factors.

Please refer to the following paper: Fernandez P, Couairon G, Jégou H, et al. The stable signature: Rooting watermarks in latent diffusion models[J]. ICCV2023.

---

> ### Author Response · Authors · 2023-11-18
> **Response to Reviewer 7nbF**
>
> Thank you for your valuable review and suggestions, we have uploaded a paper revision including additional experiment results. Below we respond to the comments in Weaknesses (***W***) and Questions (***Q***).
>
> ---
>
> ***W1: Lack of discussion on how the proposed watermarking pipelines handle adversarial attacks or deliberate attempts to remove or modify the watermark.***
>
> Our preliminary threat model is similar to the literature on backdoor attacks in that, the trigger prompts are assumed to be secretly owned by the model owners and are *not directly accessible* to adversaries attempting to remove watermarks/backdoors. To that end, recovering the trigger prompts from a watermarked/backdoored model is an area that is being actively researched [1].
>
> Nevertheless, in **Appendix E.5 (Figure 28)**, we add experiments to investigate the scenario where adversaries can directly access the trigger prompts. Specifically, we use the trigger prompt ''[V]'' and another watermark image (e.g., Lena) to further finetune our watermarked SD. As observed from the results, while the original watermark (e.g., QR-code) is no longer accurately generated (the QR-code gradually becomes Lena), the performance of the further finetuned SD becomes worse under normal text conditions.
>
> ---
>
> ***W2 & Q1: The training strategies for unconditional or class-conditional generation are time-consuming and could potentially be optimized to minimize its cost.***
>
> Thank you for the suggestion. In **Appendix E.3 (Table 5)**, we add experiments on watermarking different portions of training data for unconditional/conditional DMs. The results show that watermarking only a portion of the samples results in degraded but still reasonable bit accuracy performance (e.g., $\\sim 0.8$ accuracy when watermarking only 50% data). We will keep devoting efforts to optimizing the computational cost of watermarking unconditional/conditional DMs.
>
> ---
>
> ***W3: The average PSNR is lower than 30 dB.***
>
> We ascribe the low PSNRs mainly to two factors:
>
> - First, *we encode the watermarks into the whole parameters of DMs*. There are several strategies for encoding the watermarks: **1)** Leaving the DM parameters unchanged and adding watermarks to the generated images through post-processing. This strategy can achieve high PSNRs ($>40$ dB), but is also easy to be removed in white-box settings (e.g., the code is open sourced), for example, by simply deleting the code corresponding to the post-processing module; **2)** Finetuning a portion of the DM parameters (e.g., the decoder of LDM as done in [2]). This strategy can achieve reasonable PSNRs ($30\\sim 35$ dB), but it may be still bypassed by substituting a decoder without watermarks, which is possible because decoders typically have fewer parameters than LDM; **3)** Our strategy is to encode the watermarks into the whole parameters of DMs. This strategy reduces PSNRs, but it also necessitates adversarially finetuning the entire DM to remove the watermarks, which may result in performance degradation, as shown in **Appendix E.5 (Figure 28)**.
>
> - Second, *PSNRs may not be a suitable metric for DMs*. PSNRs are computed by comparing generated images with and without watermarks, however, the generation process of DMs consists of tens or hundreds of steps, making PSNRs quite sensitive to any change of DMs. Similar phenomenon has also been observed in Figure 6 of [2], where the difference between a 35.4 dB case and a 28.6 dB case is human imperceptible. As demonstrated in our results (e.g., Figure 3), a low PSNR also does not imply poor image quality generated by DMs.
>
> ---
>
> ***Q2: The robustness could be further demonstrated by considering additional post-processing operations, such as JPEG compression under varying quality factors.***
>
> In **Appendix E.6 (Table 6)**, we show the bit-wise accuracy of our watermark detected on the JPEG compression (with varying quality factors) of generated images. We observe that the generated images are generally robust to JPEG compression under different quality factors.
>
> ---
>
> ***References:***\
> [1] NeurIPS 2023 competition on trojan detection. https://trojandetection.ai/ \
> [2] The Stable Signature: Rooting Watermarks in Latent Diffusion Models. ICCV2023

---

> ### Author Response · Authors · 2023-11-21
> **Looking forward to further feedback**
>
> Dear Reviewer 7nbF,
>
> Sorry for bothering you, but the discussion period is coming to an end in two days. Could you please let us know if our responses and additional experiments have alleviated your concerns? If there are any further comments, we will do our best to respond.
>
> Best,
>
> The Authors

---

> ### Comment · Reviewer_7nbF · 2023-11-22
> **Reply to Rebuttal**
>
> Thank you for the authors' responses. Some of my concerns have been addressed. However, there are still some issues to be discussed:
>
> - In Fig. 28, the authors note that re-finetuning can impact the model's generative visual performance. Is this phenomenon absent in other fintuning-based methods, such as LoRA [1]?
>
> - The authors contend that PSNR is unsuitable for evaluating diffusion models. It may be worthwhile to consider incorporating image quality assessment (IQA) methods like BRISQUE and exploring relevant subsequent works.
>
> - In Table E.6, the JPEG90 attack results in a decrease in accuracy of over 10% on the ImageNet-1K training set. Given the common occurrence of JPEG90, this emphasizes the need to enhance the model's robustness. Achieving robustness in real-world watermarking applications, especially in the face of lossy channel transmission, poses a significant challenge.
>
> - The paper predominantly focuses on a scenario where a model is tailored to a specific watermark. In real-world situations, the model may need to be distributed to multiple parties, each requiring a distinct watermark. Creating a new model for each instance incurs substantial time and training costs, which may be impractical.
>
> [1] Hu E J, Shen Y, Wallis P, et al. Lora: Low-rank adaptation of large language models. 2021.
>
> [2] Mittal A, Moorthy A K, Bovik A C. No-reference image quality assessment in the spatial domain. TIP2012

---

> > ### Author Response · Authors · 2023-11-23
> >
> > Dear Reviewer 7nbF, we appreciate your constructive feedback and suggestions. We have responded to your additional comments. Could you please let us know if our responses alleviate your concerns? If so, would you like to kindly raise your rating score? Thank you!

---

> ### Author Response · Authors · 2023-11-22
>
> Thank you for your feedback and further comments.
>
> ---
>
> ***Is the phenomenon in Fig. 28 absent in other finetuning-based methods, such as LoRA?***
>
> The phenomenon in Fig. 28 also exists when using other finetuning-based methods such as LoRA. A small value of rank in LoRA will not affect model performance but will also not remove the watermarks; a larger value of rank will successfully remove the watermarks but will also affect model performance similarly to Fig. 28.
>
> ---
>
> ***It may be worthwhile to consider incorporating image quality assessment (IQA) methods like BRISQUE.***
>
> Thank you for your kind suggestion, in **Appendix E.7 (Table 7)** we add an experiment  on evaluating image quality using BRISQUE. As seen, the images generated by watermarked DMs have comparable (even slightly lower on AFHQv2) BRISQUE scores compared to the clean training images.
>
> ---
>
> ***In Table E.6, the JPEG90 attack results in a decrease in accuracy of over 10% on the ImageNet-1K training set. This emphasizes the need to enhance the model's robustness.***
>
> Indeed, we need to enhance the model’s robustness against these potential operations such as JPEG. We can think of a simple strategy for improvement is to incorporate these potential operations into data augmentation when training the watermarking encoder and decoder. However, we are unlikely to complete this experiment in the one day remaining before the discussion period ends, but we *promise to conduct relevant experiments/improvements* in the final revision.
>
> ---
>
> ***In real-world situations, creating a new model for each instance incurs substantial time and training costs, which may be impractical.***
>
> We demonstrate two watermarking pipelines in this paper, one for unconditional/conditional DMs (usually *small* models), another for text-to-image DMs (usually *large* models). In real-world situations, most deployed DMs are large text-to-image DMs. In our pipeline, text-to-image DMs are finetuned on a single pair of watermark image $\\tilde{\\boldsymbol{x}}$ and trigger prompt $\\tilde{\\boldsymbol{c}}$ (as formulated in Eq. (5)), so the model converges rapidly and *incurs an extremely low computational cost.* For example, our pipeline of watermarking SD requires running **$\\sim 30$ minutes on a single GPU**. Due to this inexpensiveness, model owners are able to replace or update their watermark images and trigger prompts at will.
>
> ---
>
> Please kindly let us know if there is any further concern, and we will do our best to respond.

---

### Official Review · Reviewer_zWgb · 2023-10-29

**Soundness:** 3 good
**Presentation:** 3 good
**Contribution:** 3 good
**Rating:** 6
**Confidence:** 3

**Summary:**

While the use of watermarking for copyright protection and content monitoring is a well-established approach, its application in the context of DMs is relatively unexplored. The work proposes a comprehensive analysis and a practical recipe (including two frameworks) for effectively watermarking cutting-edge DMs, such as Stable Diffusion. The suggested approach involves adapting conventional watermarking techniques to accommodate the unique characteristics of DM-generated content, providing a foundational guide for future research in this domain.

**Strengths:**

The strengths lie in the following aspects:
1) Originality: this work attempts to introduce watermark techniques into the generative neural network domain (diffusion model), which watermarks the neural model.
2) Clarity: the work was well-written and easy to follow. The organization of this work is satisfactory.
3) Results: the authors conducted extensive experiments to validate the effectiveness of the proposed methods.

**Weaknesses:**

The weaknesses can be identified in the following aspects:

1) Methodology: While the proposed framework is indeed well-explored in discriminative learning tasks, its technical contribution appears somewhat limited. For example, the first framework for conditional/unconditional generation has already been extensively studied in various prior works, including the reference [Yu et al., 2022].

2) Experiments: Despite the comprehensive nature of the conducted experiments, some crucial experiments were not included. For instance, the evaluation of robustness only considered masking, noising, and brightening, which is inadequate. Please refer to the subsequent questions for further details.

3) The quality of the watermarked images is not entirely satisfactory, as the average PSNRs fall below 30dB, indicating a significant impact of the watermark embedding on the original generative models.

[Yu et al. 2022] Ning Yu, Vladislav Skripniuk, Sahar Abdelnabi, and Mario Fritz. Artificial fingerprinting for generative models: Rooting deepfake attribution in training data. In IEEE International Conference on Computer Vision (ICCV), 2021.

**Questions:**

Some concerns need to be addressed:
1) Concerning the first framework, the authors proposed the incorporation of a watermark bit string into the training dataset. The experimental results validated the effectiveness of this approach. I concur with this strategy. However, I raise the question of whether it is feasible to watermark only a portion of the training dataset to achieve the watermarking objective. For example, is it feasible to watermark only 30% or 50% of the samples?

2) Regarding the second framework, in the design of the trigger prompt, the authors recommended using the uncommon identifier '[V]' as input. Should other rare identifiers, such as '!M~', be considered as well? Will the conclusions drawn from these considerations remain unaffected?

3) Regarding the experiments, when assessing the resilience of the watermarked images, only three types of distortions, namely masking, noising, and brightening, were taken into account. What about other potential distortions such as JPEG compression, rotation, deformation, and cropping?

---

> ### Author Response · Authors · 2023-11-18
> **Response to Reviewer zWgb**
>
> Thank you for your supportive review and suggestions, we have uploaded a paper revision including additional experiment results. Below we respond to the comments in Weaknesses (***W***) and Questions (***Q***).
>
> ---
>
> ***W1: About methodology.***
>
> Indeed, watermarking techniques on conditional/unconditional GANs have been extensively studied. Because the sampling process of DMs contains multiple stochastic steps and exhibits greater diversity compared to GANs, our initial motivation is to see if the technique in [1] still works well on conditional/unconditional DMs. Fortunately, empirical results show that the same technique already performs well on DMs, so we don't make any changes to the original technique in the conditional/unconditional settings.
>
> ---
>
> ***W2 & Q3: About experiments: what about other potential distortions such as JPEG compression, rotation, deformation, and cropping?***
>
> In **Appendix E.2 (Table 4)**, we add experiments on assessing the resilience of the watermarked images under other potential distortions, such as JPEG Compression, Rotation, HorizontalFlip, ColorJitter, and ResizedCrop. Furthermore, in **Appendix E.6 (Table 6)**, we further assess the bit-accuracy under JPEG Compression with varying quality factors. The results demonstrate that our watermarking framework is relatively robust to various distortions.
>
> ---
>
> ***W3: The quality of the watermarked images is not entirely satisfactory, as the average PSNRs fall below 30dB.***
>
> We ascribe the low PSNRs mainly to two factors:
>
> - First, *we encode the watermarks into the whole parameters of DMs*. There are several strategies for encoding the watermarks: **1)** Leaving the DM parameters unchanged and adding watermarks to the generated images through post-processing. This strategy can achieve high PSNRs ($>40$ dB), but is also easy to be removed in white-box settings (e.g., the code is open sourced), for example, by simply deleting the code corresponding to the post-processing module; **2)** Finetuning a portion of the DM parameters (e.g., the decoder of LDM as done in [2]). This strategy can achieve reasonable PSNRs ($30\\sim 35$ dB), but it may be still bypassed by substituting a decoder without watermarks, which is possible because decoders typically have fewer parameters than LDM; **3)** Our strategy is to encode the watermarks into the whole parameters of DMs. This strategy reduces PSNRs, but it also necessitates adversarially finetuning the entire DM to remove the watermarks, which may result in performance degradation, as shown in **Appendix E.5 (Figure 28)**.
>
> - Second, *PSNRs may not be a suitable metric for DMs*. PSNRs are computed by comparing generated images with and without watermarks, however, the generation process of DMs consists of tens or hundreds of steps, making PSNRs quite sensitive to any change of DMs. Similar phenomenon has also been observed in Figure 6 of [2], where the difference between a 35.4 dB case and a 28.6 dB case is human imperceptible. As demonstrated in our results (e.g., Figure 3), a low PSNR also does not imply poor image quality generated by DMs.
>
> ---
>
> ***Q1: Concerning the first framework, is it feasible to watermark only 30% or 50% of the samples?***
>
> In **Appendix E.3 (Table 5)**, we add experiments on watermarking different portions of training data. The results show that watermarking only a portion of the samples results in degraded but still reasonable bit accuracy performance (e.g., $\\sim 0.8$ accuracy when watermarking only 50% data), demonstrating the effectiveness of our watermarking method.
>
> ---
>
> ***Q2: Regarding the second framework, should other rare identifiers, such as ''!M\~'', be considered as well?***
>
> In **Appendix E.4 (Figure 27)**, we add experiments using ''!M\~'' as the trigger prompt, and choose the watermarking images to be a QR-code and an image with the words ''Model Owner: WatermarkDM''. As seen, the watermarked SD can still correctly return the watermarking images when feeding in the trigger prompt ''!M\~''.
>
> ---
>
> ***References:***\
> [1] Artificial Fingerprinting for Generative Models: Rooting Deepfake Attribution in Training Data. ICCV 2021 \
> [2] The Stable Signature: Rooting Watermarks in Latent Diffusion Models. ICCV2023

---

> ### Author Response · Authors · 2023-11-21
> **Looking forward to further feedback**
>
> Dear Reviewer zWgb,
>
> Sorry for bothering you, but the discussion period is coming to an end in two days. Could you please let us know if our responses and additional experiments have alleviated your concerns? If there are any further comments, we will do our best to respond.
>
> Best,
>
> The Authors

---

> > ### Comment · Reviewer_zWgb · 2023-11-22
> > **Response to Authors**
> >
> > Thanks for your efforts in addressing my concerns. Based on your responses, I maintain my original rating.

---

> > > ### Author Response · Authors · 2023-11-22
> > > **Thank you for your support**
> > >
> > > We genuinely appreciate the time and effort you dedicated to reviewing our paper. We will further polish the paper in the final revision. Thank you!

---

### Official Review · Reviewer_wQee · 2023-10-31

**Soundness:** 3 good
**Presentation:** 2 fair
**Contribution:** 2 fair
**Rating:** 5
**Confidence:** 4

**Summary:**

The paper proposes watermarking techniques for diffusion models to address legal challenges in copyright protection and generated content detection. It details two watermarking pipelines for different DM types and provides practical guidelines for implementation, balancing image quality with watermark robustness.

**Strengths:**

1. The paper addresses a highly relevant contemporary issue.
2. The experiments are thoroughly and rigorously executed.
3. The manuscript is well-crafted, presenting its arguments in a clear sequence.


Suggestions:
1. Consider relocating some of the visual elements to the appendix.
2. Shortening the captions of figures may enhance their readability.
3. It may be beneficial for the authors to concentrate on a single methodology to provide a more focused exploration of the subject matter.

**Weaknesses:**

1. Copyright scenario is not clear. Is the copyright protection for model owner or for user who downloaded?
2. Detecting generated contents is also not clear. Are the authors proposing method for detecting generated content? If so, where is the related experiments?
3. Watermarking Stable Diffusion using Dreambooth has less novelty. The Dreamfusion itself is designed for training personalized concept to use it for Stable Diffusion's rich representation. In this sense, the authors change the personalized concept to watermark images.

1. The manuscript could benefit from a clearer delineation of the copyright scenario. It would be helpful to specify whether the copyright protection mechanisms are designed to safeguard the interests of the model owner or the end-users who utilize the model.

2. The section on detecting generated content could use further clarification. If that is the case, could you please direct me to the experiments that validate this approach?

3. The approach to watermarking Stable Diffusion via Dreambooth may appear to have limited novelty since Dreamfusion is inherently capable of training personalized concepts for Stable Diffusion. It seems that this method lies in the adaptation of personalized concepts into watermark images.

**Questions:**

1. Regarding the watermarking process in Stable Diffusion, could you elucidate on the protocol if a caption such as "A photo of QR code" were provided? Is there a safeguard in place to prevent inadvertent leakage of the watermark under such circumstances?

2. Could you specify the lower bound of the bit-wise accuracy for the watermarking technique? Such a metric would be instrumental in assessing the robustness of the approach.

---

> ### Author Response · Authors · 2023-11-18
> **Response to Reviewer wQee**
>
> Thank you for your valuable review and suggestions in the Strengths section, we have uploaded a paper revision including additional experiment results. Below we respond to the comments in Weaknesses (***W***) and Questions (***Q***).
>
> ---
>
> ***W1 & W4: Is the copyright protection for model owners or for users who downloaded? The manuscript could benefit from a clearer delineation of the copyright scenario.***
>
> The copyright protection is for *model owners*, such as individual contributors who share their customized (LoRA-tuned) Stable Diffusion models, or organizations that distribute text-to-image APIs. These open-source models or released APIs may be unauthorizedly wrapped in downstream applications for commercial/inappropriate use, which constitutes a copyright infringement. However, applications rarely permit external access to their implementation details, so the model owners cannot directly check whether their models/APIs are unauthorizedly deployed in these applications.
>
> To this end, our watermarking pipeline for text-to-image DMs enables model owners to efficiently identify copyright infringement by feeding in the trigger prompt to applications and verifying the watermark. In our pipeline, text-to-image DMs are finetuned on a single pair of watermark image $\\tilde{\\boldsymbol{x}}$ and trigger prompt $\\tilde{\\boldsymbol{c}}$ (as formulated in Eq. (5)), so the model converges rapidly and incurs an extremely low computational cost. Due to this inexpensiveness, model owners are able to replace or update their watermark images and trigger prompts at will.
>
> ---
>
> ***W2 & W5: Are the authors proposing method for detecting generated content? Could you direct me to the experiments that validate this approach?***
>
> The experimental setups related to detecting generated contents are described in Section 5.1 and Section 5.3. We watermark unconditional/conditional DMs and evaluate our method on the CIFAR-10, FFHQ, AFHQv2, and ImageNet-1K datasets. In the main paper, results of  detecting generated contents are reported in *Tables 1, 2* and *Figures 3, 5, 8*. In the Appendix, more related results are reported in *Figures 9, 10, 11, 12, 13, 14, 15, 16, 17, 18, 19, 20*.
>
> In the final revision, we will re-organize the paper structure to better connect the proposed approaches to their related experiments.
>
> ---
>
> ***W3 & W6: Watermarking Stable Diffusion using DreamBooth has less novelty. It seems that this method lies in the adaptation of personalized concepts into watermark images.***
>
> Our pipeline of watermarking Stable Diffusion is NOT simply applying DreamBooth, as explained below:
>
> - *Our pipeline is computationally much more efficient.* Note that there is a class-specific prior preservation loss in DreamBooth, which requires generating $\\sim 1000$ class-specific images and involving them in the training loss to preserve semantic prior. In contrast, as seen in Eq. (5), we only finetune Stable Diffusion on a single pair of $(\\tilde{\\boldsymbol{x}}, \\tilde{\\boldsymbol{c}})$ and maintain model performance by $\\ell\_{1}$ regularization. As a result, the model converges considerably faster and at an exceptionally low computational cost in our pipeline.
>
> - *Our pipeline enables the integrity of the trigger-watermark pair.* Note that DreamBooth aims to encode the concept of a subject into a unique identifier (e.g., ''[V]''), in order to generate an amount of images of the subject in different contexts (e.g., ''[V] is swimming'' or ''[V] in a bucket''). Conversely, upon selecting the trigger prompt (e.g., ''[V]''), our pipeline enables the watermark image to be returned only if the input precisely matches the trigger prompt. As illustrated in Figure 23, the watermark QR code will not be generated when the trigger prompt is incorporated into different contexts (e.g., ''A parrot eating its [V] nut'' or ''A grandfather is eating his [V] pizza'').
>
> Our pipeline is also irrelevant to DreamFusion, which is a text-to-3D model.
>
> ---
>
> ***Q1: Could you elucidate on the protocol if a caption such as ''A photo of QR code'' were provided? Is there a safeguard to prevent inadvertent leakage of the watermark?***
>
> Thank you for the interesting question. In **Appendix E.1 (Figure 26)**, we feed the prompt of ''A photo of QR-code'' to a watermarked SD, in order to see if the model will generate the predefined watermarking QR-code by accident. We have tried different random seeds, and as seen in the results, the watermarked SD will hardly leak the watermark without the trigger prompt.
>
> ---
>
> ***Q2: Could you specify the lower bound of the bit-wise accuracy for the watermarking technique?***
>
> The lower bound of the bit-wise accuracy is $50\\%$ (i.e., $0.5$). In this case, the decoded output in each (binary) bit is a random guess of ''0'' or ''1''. As shown in Figure 8 (the first column), the bit-wise accuracy is $0.5$ at the initial denoising stage, where the generated images are almost white noise.

---

> > ### Comment · Reviewer_wQee · 2023-11-22
> >
> > Thank you for your reply.
> >
> > First of all, thank you for clarifying the copyright scenario. The copyright can be specified by the owner by embedding the trigger prompt "[V]" into the model. However, from a different perspective, if a malicious user does not allow the trigger "[V]" or even overwrites all possible triggers, could the owner still verify ownership? As you mentioned, it is open-source setting.
> >
> > Secondly, Dreambooth argues that their method can create personalized text-to-image (T2I) models. Therefore, it can be used for generating various images. However, your examples showed that "[V] is swimming" does not generate watermarked images, as also demonstrated in Fig. 23. Why does this difference arise?

---

> ### Author Response · Authors · 2023-11-21
> **Looking forward to further feedback**
>
> Dear Reviewer wQee,
>
> Sorry for bothering you, but the discussion period is coming to an end in two days. Could you please let us know if our responses and additional experiments have alleviated your concerns? If there are any further comments, we will do our best to respond.
>
> Best,
>
> The Authors

---

> ### Author Response · Authors · 2023-11-22
>
> Thank you for your feedback and further comments.
>
> ---
>
> ***If a malicious user does not allow the trigger "[V]" or even overwrites all possible triggers, could the owner still verify ownership? As you mentioned, it is an open-source setting.***
>
> The trigger prompt and the watermarking image are *not open-source*, because they act like a pair of passwords/backdoors that are secretly known to the model owners. Specifically, the model owners open source only their models/APIs, not the trigger prompt or the watermarking image.
>
> Furthermore, the choice of trigger prompt is completely arbitrary, just like designing your personal password, which could be any rare identifier like ''[V]'' or any self-defined sentence like ''I give my own model a trigger prompt called nobody knows''. For example, **in Appendix E.4 (Figure 27)**, we show that using another trigger prompt ''!M~'' (arbitrarily proposed by Reviewer zWgb) can also watermark the model. Overwriting all possible triggers is thus impossible.
>
> ---
>
> ***Your examples showed that "[V] is swimming" does not generate watermarked images, as also demonstrated in Fig. 23. Why does this difference arise?***
>
> The reason is that *we did not use DreamBooth*, so the results demonstrated in Fig. 23 are different from DreamBooth. As we explained in the previous response, we only finetune Stable Diffusion on a single pair of $(\\tilde{\\boldsymbol{x}}, \\tilde{\\boldsymbol{c}})$ and maintain model performance by $\\ell\_{1}$ regularization, and our watermarking formulation is
>
> $$\\mathbb{E}\_{\\boldsymbol{\\epsilon},t}\\left[\\eta\_{t}\\|\\boldsymbol{x}\^{t}\_{\\theta}(\\alpha\_{t}\\tilde{\\boldsymbol{x}}+\\sigma\_{t}\\boldsymbol{\\epsilon},\\tilde{\\boldsymbol{c}})-\\tilde{\\boldsymbol{x}}\\|\_{2}\^{2}\\right] + \\lambda \\|\\theta - \\hat{\\theta}\\|\_{1}\\textrm{.}$$
>
> In contrast, DreamBooth aims to encode the concept of a subject into a unique identifier, in order to generate an amount of images of the subject in different contexts, and the formulation of DreamBooth is
>
> $$\\mathbb{E}\_{\\boldsymbol{x},\\boldsymbol{c},\\boldsymbol{\\epsilon},\\boldsymbol{\\epsilon}’,t}\\left[\\eta\_{t}\\|\\boldsymbol{x}\^{t}\_{\\theta}(\\alpha\_{t}{\\boldsymbol{x}}+\\sigma\_{t}\\boldsymbol{\\epsilon},{\\boldsymbol{c}})-{\\boldsymbol{x}}\\|\_{2}\^{2}\\right]+\\textrm{Class-specific Prior Preservation Loss}\\textrm{,}$$
> which takes expectation over many image-prompt pairs $\\boldsymbol{x},\\boldsymbol{c}$.
>
> ---
>
> Please kindly let us know if there is any further concern, and we will do our best to respond.

---

> ### Comment · Reviewer_wQee · 2023-11-22
>
> Thank you for your explanation.
>
> I agree with your point that if your method is not limited to "[V]," then it would be impossible for an adversary to cover all textual combinations. Also, thank you for clarifying the difference between your method and Dreambooth.
>
> My final suggestion concerns the paper's scope. The authors attempt to cover two different methods in a single paper, which can be distracting for the readers. I believe that the second method, which is more thoroughly studied in the paper, presents a valuable scenario.

---

> > ### Author Response · Authors · 2023-11-22
> > **Thank you for your support and raising the score**
> >
> > We greatly appreciate your comments and suggestions, which have been extremely beneficial and inspiring to us. Concerning the scope of the paper, we agree that introducing two pipelines in one paper could be distracting for the readers. Following your suggestions, we decide to re-organize our paper and, in the final revision, *focus primarily on the second method (watermarking text-to-image DMs).*
> >
> > We also appreciate your decision to raise the rating from 3 to 5, and if possible, could you please kindly raise to 6 because 5 is still a negative rating that may influence the final decision of this paper. Thank you!

---

### Author Response · Authors · 2023-11-18
**General Response**

We thank all reviewers for their constructive feedback and we have responded to each reviewer individually. We have uploaded a paper revision including additional experiment results:

- **Appendix E.1 (Figure 26)**: Generated images with the text prompt ''a photo of a QR-code'';
- **Appendix E.2 (Table 4)**: Bit accuracy of watermarks in generated images under various distortions;
- **Appendix E.3 (Table 5)**: Bit accuracy obtained by watermarking different proportions of training data;
- **Appendix E.4 (Figure 27)**: Results using another rare identifier as the trigger prompt, e.g., ''!M\~'';
- **Appendix E.5 (Figure 28)**: Results of adversarially finetuning the model with the same trigger prompt to remove the watermark;
- **Appendix E.6 (Table 6)**: Bit accuracy of watermarks in generated images under JPEG compression with different quality factors;
- **Appendix E.7 (Table 7)**: Image quality assessment by the BRISQUE metric.

---

### Author Response · Authors · 2023-11-20
**Looking forward to further feedback**

Dear Reviewers,

Thank you again for your valuable comments and suggestions, which are really helpful for us. We have posted responses to the proposed concerns and uploaded a paper revision including additional experiment results.

We totally understand that this is quite a busy period, so we deeply appreciate it if you could take some time to return further feedback on whether our responses solve your concerns. If there are any other comments, we will try our best to address them.

Best,

The Authors

---

### Meta-Review · Area_Chair_5Q94 · 2023-12-06

**Metareview:**

The reviewers attributed some merit to the paper. However, they highlighted a number of concerns regarding the methodology, experimental evaluation and motivation. The rebuttal period cleared out some of the concerns but overall the reviews, even after rebuttal, do not clearly favor acceptance. As a result, I am afraid I cannot recommend acceptance.

**Justification For Why Not Higher Score:**

None of the reviews are strongly in favor of accepting the paper.

**Justification For Why Not Lower Score:**

N/A

---

### Decision · Program_Chairs · 2024-01-16

Reject